# Selective clearance of aberrant tau proteins and rescue of neurotoxicity by transcription factor EB

Vinicia A Polito[1,†], Hongmei Li[1,†], Heidi Martini-Stoica[1,2,3,†], Baiping Wang[1,4], Li Yang[1], Yin Xu[1], Daniel B Swartzlander[1], Michela Palmieri[4,5], Alberto di Ronza[4,5], Virginia M-Y Lee[6], Marco Sardiello[4,5], Andrea Ballabio[4,5,7] & Hui Zheng[1,3,4,*]

## Abstract

Accumulating evidence implicates impairment of the autophagy-lysosome pathway in Alzheimer's disease (AD). Recently discovered, transcription factor EB (TFEB) is a molecule shown to play central roles in cellular degradative processes. Here we investigate the role of TFEB in AD mouse models. In this study, we demonstrate that TFEB effectively reduces neurofibrillary tangle pathology and rescues behavioral and synaptic deficits and neurodegeneration in the rTg4510 mouse model of tauopathy with no detectable adverse effects when expressed in wild-type mice. TFEB specifically targets hyperphosphorylated and misfolded Tau species present in both soluble and aggregated fractions while leaving normal Tau intact. We provide *in vitro* evidence that this effect requires lysosomal activity and we identify phosphatase and tensin homolog (PTEN) as a direct target of TFEB that is required for TFEB-dependent aberrant Tau clearance. The specificity and efficacy of TFEB in mediating the clearance of toxic Tau species makes it an attractive therapeutic target for treating diseases of tauopathy including AD.

**Keywords** Alzheimer's disease; tauopathy; TFEB; PTEN; autophagy-lysosomal pathway
**Subject Categories** Neuroscience

## Introduction

Alzheimer's disease (AD) is characterized by the presence of extracellular amyloid plaques consisting of β-amyloid peptides (Aβ) and intracellular neurofibrillary tangles (NFT) composed of hyperphosphorylated Tau (pTau) protein in diseased brains. Biochemical and genetic studies of the amyloid precursor protein (APP) and presenilins support a causal role of Aβ in AD pathogenesis (reviewed by (Hardy, 2006)). Accordingly, Aβ-based therapies have been actively pursued. However, the clinical outcomes of these therapies have been disappointing so far (reviewed in (Mullard, 2012)). Of note, these agents may not target the NFT pathology and strong evidence supports a direct role of misfolded Tau and NFTs in AD and other neurodegenerative diseases (reviewed by (Gendron & Petrucelli, 2009; Mandelkow & Mandelkow, 2012)). Tau, encoded by *MAPT*, is typically localized to axons where it binds and stabilizes microtubules. Aberrant Tau misfolding due to hyperphosphorylation or other alterations leads to its dissociation from microtubules followed by aggregation and redistribution to cell bodies and dendrites. Clinically, NFT pathology correlates with dementia better than amyloid plaques (Giannakopoulos *et al*, 2003). Although no Tau mutations have been found in AD, mutations in the *MAPT* gene are causal for a subtype of frontotemporal dementia (FTD), termed FTD with Parkinsonism linked to chromosome 17 (FTDP-17). These mutations are known to impair Tau structure and promote its fibrillization (Gendron & Petrucelli, 2009; Mandelkow & Mandelkow, 2012). Experimentally, overexpression of FTD-associated *MAPT* mutant genes in transgenic mice results in NFT development and neurodegeneration (Ramsden *et al*, 2005; Santacruz *et al*, 2005), establishing the neurotoxicity conferred by the mutant Tau proteins. As such, there is increasing interest in developing Tau-based therapy for treating diseases of tauopathy including AD and FTD (reviewed in (Brunden *et al*, 2009)).

Macroautophagy (herein referred to as autophagy) is a conserved mechanism that cells utilize to degrade intracellular long-lived proteins and organelles through lysosome-mediated degradation. Accumulating evidence has implicated an impaired autophagy and lysosomal pathway (ALP) in neurodegenerative diseases (see (Harris & Rubinsztein, 2012; Nixon & Yang, 2011) for recent reviews).

1  Huffington Center on Aging, Baylor College of Medicine, Houston, TX, USA
2  Interdepartmental Program of Translational Biology and Molecular Medicine, Baylor College of Medicine, Houston, TX, USA
3  Medical Scientist Training Program, Baylor College of Medicine, Houston, TX, USA
4  Department of Molecular and Human Genetics, Baylor College of Medicine, Houston, TX, USA
5  Dan and Jan Duncan Neurological Research Institute, Texas Children's Hospital, Houston, TX, USA
6  Department of Pathology and Lab Medicine, University of Pennsylvania School of Medicine, Philadelphia, PA, USA
7  Department of Translational Medical Sciences, Section of Pediatrics, Telethon Institute of Genetics and Medicine, Federico II University, Naples, Italy
*Corresponding author. Tel: +1 713 798 1568; Fax: +1 713 798 1610; E-mail: huiz@bcm.edu
†These authors contributed equally to this work

Specific to AD, the ALP has been documented to regulate APP turnover and Aβ metabolism (Mueller-Steiner *et al*, 2006; Nixon, 2007; Pickford *et al*, 2008; Rohn *et al*, 2011; Yang *et al*, 2011) as well as Tau protein degradation (Wang *et al*, 2009, 2010; Kruger *et al*, 2012; Schaeffer *et al*, 2012; Caccamo *et al*, 2013; Ozcelik *et al*, 2013). A physiological role of autophagy in Tau homeostasis is demonstrated through pTau accumulation and neurodegeneration in mice with neuronal deletion of the autophagy gene *Atg7* (Inoue *et al*, 2012).

Autophagy is mediated by a series of intracellular membrane trafficking events and is executed by lysosomal degradation of sequestered contents. Thus, efficient autophagy requires heightened lysosomal activity. The Transcription Factor EB (TFEB) was recently discovered as a master regulator of the ALP through coordinated expression of autophagy and lysosomal target genes and enhanced lysosomal biogenesis (Sardiello *et al*, 2009; Settembre *et al*, 2011). TFEB is normally sequestered in the cytoplasm by phosphorylation, an event shown to be mediated by the key negative autophagy regulator mTOR (Roczniak-Ferguson *et al*, 2012; Settembre *et al*, 2012). Accordingly, mTOR inhibition is associated with TFEB dephosphorylation and nuclear translocation, leading to the induction of downstream targets by binding to the coordinated lysosomal expression and regulation (CLEAR) element (Sardiello *et al*, 2009; Settembre *et al*, 2011). Here we investigated the role of TFEB in APP and Tau transgenic mouse models and found that TFEB potently cleared pTau/NFT pathologies and rescued neurodegenerative and behavioral deficits without overtly affecting Aβ pathology or exhibiting adverse effects in wild-type mice.

# Results

## TFEB differentially targets Aβ and pTau/NFT pathologies

We used an adeno-associated virus (AAV) delivery approach to assess the potential effect of TFEB in 5xFAD (Oakley *et al*, 2006) and rTg4510 (Ramsden *et al*, 2005; Santacruz *et al*, 2005) transgenic mouse models, which develop progressive Aβ and NFT neuropathologies, respectively, starting at approximately 2 months of age. The AAV2/9 vector containing mouse *TFEB* cDNA (AAV-TFEB) or *GFP* (AAV-GFP) driven by the CMV promoter was injected into the lateral ventricles of both cerebral hemispheres on postnatal day 0 (P0) of 5xFAD or rTg4510 mouse brains and their wild-type littermate controls. Assessment of mice injected with AAV-GFP revealed widespread brain expression (Supplementary Fig S1A), particularly in cortical and hippocampal neurons (Supplementary Fig S1B). Quantitative real-time PCR (qRT-PCR) analysis of forebrain samples 1-month post-injection showed approximately a twofold to threefold elevation of *TFEB* expression compared to uninjected or GFP-injected controls (Supplementary Fig S1C). Long-lasting TFEB expression is evidenced by similar levels of TFEB overexpression when analyzed 4-month post-injection (see Fig 1D and F).

The 5xFAD and rTg4510 mice were analyzed 4 months after TFEB injection. Western blot analysis revealed no appreciable changes of APP levels in TFEB-injected wild-type or 5xFAD mice as compared to uninjected controls (Supplementary Fig S2A and B). Immunohistochemical staining and quantification of Aβ pathology also showed that TFEB had no appreciable effect on amyloid deposition (Supplementary Fig S2C–F).

In sharp contrast, using antibodies against phospho-Tau (AT8, S202/T205 and PHF1, S396/S404) or conformation-specific Tau (MC1) species, we found that TFEB treatment drastically reduced phospho-Tau (pTau) and NFT-like pathologies in both the cortex (Fig 1A and quantified in 1B) and hippocampus (Supplementary Fig S3) of rTg4510 (Tau) mice. Reduction of MC1-positive Tau supports a role of TFEB in recognizing misfolded Tau. Western blot analysis of detergent soluble protein lysates using these antibodies as well as antibodies against total Tau and unphosphorylated Tau at the S202/T205 sites (Tau1), allowed us to determine that TFEB had no effect on wild-type Tau proteins (Fig 1C and E, WT vs. WT + TFEB); it, however, drastically reduced the AT8-, PHF1- and MC1-positive Tau species in Tau transgenic mice (Fig 1C and F, Tau vs. Tau + TFEB). Furthermore, levels of TFEB overexpression inversely correlated with that of PHF1-Tau (Fig 1D and F). Since the Tau1 levels remained constant, reduction of AT8-positive Tau most likely indicates that TFEB expression leads to the degradation rather than dephosphorylation of the phospho-Tau (pTau). However and inconsistent with this view, total Tau levels were not significantly altered (Fig 1C and quantified in 1F), suggesting that the pTau pool may represent only a small pool of total Tau levels or that TFEB might also act on Tau phosphorylation/dephosphorylation.

## TFEB promotes the clearance of aberrant Tau species

To further investigate the nature of pTau regulation by TFEB, we prepared detergent-free cytosolic and sarkosyl-insoluble fractions from Tau mouse brains with or without TFEB injection. Western blot analysis revealed that, consistent with the detergent extractable preparations, TFEB efficiently reduced the CP13- and PHF1-positive pTau while leaving Tau1-positive unphosphorylated Tau intact in both fractions (Fig 2A and quantified in B and C). Interestingly, while levels of total Tau remained similar in the soluble pool (Fig 2B), they were significantly reduced in the sarkosyl-insoluble preparations in TFEB-treated samples (Fig 2C). These results indicate that TFEB targets the detergent-insoluble pTau for degradation. This assessment is consistent with the TFEB's role in the autophagy-lysosomal pathway and is in agreement with the published reports that autophagy activators such as rapamycin or trehalose are effective in removing Tau aggregates (Schaeffer *et al*, 2012; Ozcelik *et al*, 2013).

To probe the cellular mechanisms mediating TFEB reduction of soluble pTau, we used a doxycycline inducible cell line expressing the largest human Tau isoform with the P301L mutation (T40PL). Using the same detergent extraction protocol as that used for brain lysates, we show that transfection of TFEB results in prominent reductions of total Tau and CP13- and PHF1-positive pTau without affecting unphosphorylated Tau detected by the Tau1 antibody (Fig 2D and quantified in 2E). The combined *in vitro* and *in vivo* data provides a compelling argument that the primary effect of TFEB is to target hyperphosphorylated and misfolded Tau proteins for degradation. The insignificant changes of total Tau in the soluble fraction *in vivo* may be attributed to the low percentage of pTau in the total Tau pool. However, the possibility remains that TFEB may affect Tau phosphorylation/dephosphorylation at Tau1-independent sites in the soluble pool.

    

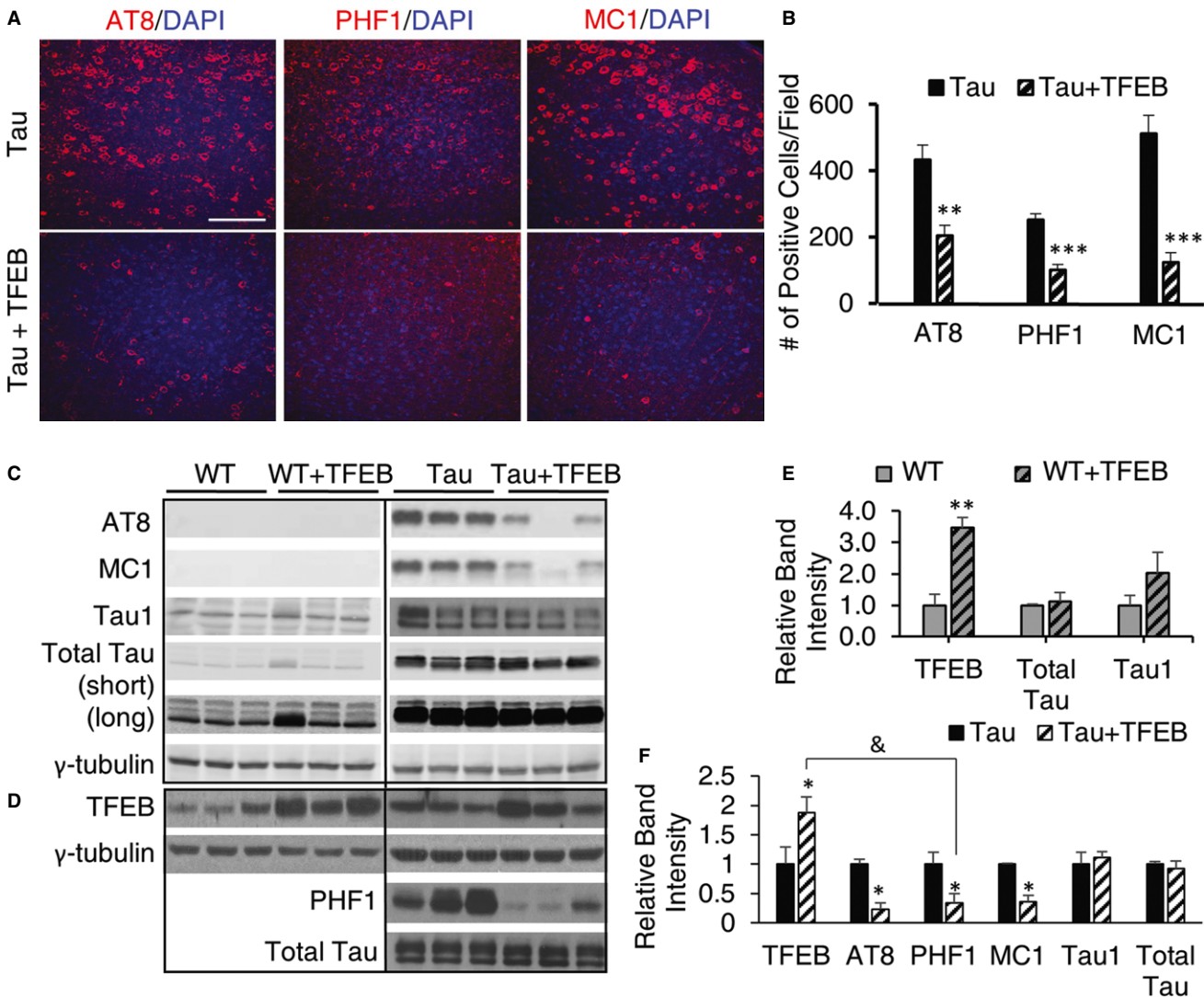

**Figure 1.  Potent reduction of pTau proteins by AAV-mediated TFEB expression.**

A   Immunofluorescence staining of cortex of 4-month-old rTg4510 mice either untreated (Tau) or injected with TFEB (Tau + TFEB) at P0 using phospho-Tau antibodies AT8 (S202/T205) and PHF1 (S396/S404) or conformation-specific antibody MC1 and counter-stained with DAPI. AAV-TFEB was injected at P0 and mice were analyzed at 4 months. Scale bar: 200 μm.

B   Quantification of staining intensities. Student's $t$-test was performed to analyze the significance. $n$ = 5 mice/genotype/treatment group. **$P$ = 0.0011, ***$P$ = 6.85 × 10$^{-5}$, and ***$P$ = 9.33 × 10$^{-5}$ for AT8, PHF1 and MC1, respectively. Each bar represents average $\pm$ s.e.m.

C   Western blot analysis of detergent-extracted brain lysates using anti-total Tau, Tau1, AT8 or MC1 antibodies. WT: wild-type; WT + TFEB: wide-type mice injected with TFEB; Tau: rTg4510 transgenic mice; Tau + TFEB: rTg4510 transgenic mice injected with TFEB. Two exposures were displayed for total Tau: the short exposure was used for quantification of Tau transgenic samples; the long exposure was used for side-by-side comparisons of the total Tau levels in WT and Tau mice and for quantification of WT total Tau. γ-tubulin was used as a loading control.

D   Western blot analysis of TFEB expression in 4-month-old WT or Tau mice with (+TFEB) or without AAV-TFEB P0 injection (the same as in A and C but from another independent batch of experiments). PHF1 and total Tau antibodies were used to blot the Tau transgenic lysates for correlation with TFEB levels.

E, F   Quantification of relative band intensities in WT (E) or Tau (F) mice. Student's $t$-test shows that TFEB protein levels are significantly increased in WT + TFEB vs WT (**$P$ = 0.0026) and in Tau + TFEB vs Tau (*$P$ = 0.047). AT8, PHF1 and MC1 levels are significantly reduced in Tau + TFEB vs Tau (*$P$ = 0.032, 0.041, 0.040, respectively). &, significant correlation between TFEB protein levels and PHF-1 was determined using Pearson Product Moment Correlation test ($P$ = 0.0044). $n$ = 3/group/experiment. Each bar represents average $\pm$ s.e.m.

Source data are available online for this figure.

## Cell autonomous effects of TFEB in pTau reduction

Having established a potent effect of TFEB in pTau reduction, we next examine whether this reduction is mediated by a cell autonomous

mechanism. We performed P0 injections of AAV-GFP or AAV-GFP/ AAV-TFEB in Tau transgenic mice. Immunostaining using the AT8 antibody revealed abundant GFP/AT8-double-positive cells in GFP-injected mice. In contrast, GFP-positive cells were devoid of pTau

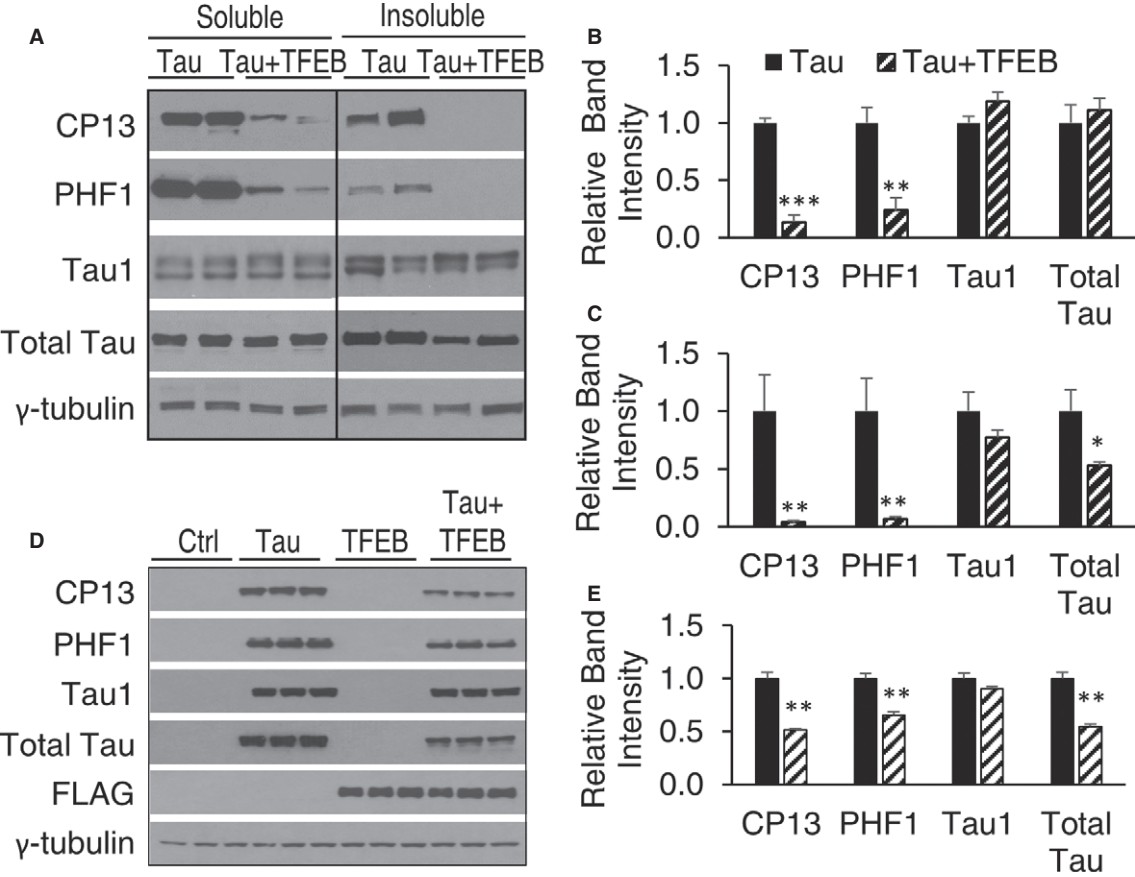

**Figure 2. Biochemical analysis of TFEB-mediated reduction of pTau.**

A    Western blot analysis of detergent-free (Soluble) and sarkosyl-insoluble (Insoluble) preparations of total (total Tau), unphosphorylated (Tau1) or CP13- (S202/T205) and PHF1-positive pTau species in 4-month-old Tau or Tau + TFEB mice with P0 injection. γ-tubulin was used as a loading control.

B, C    Quantification of relative band intensities in soluble (B) and insoluble (C) fractions. $n = 3$ mice/genotype/treatment group. CP13 and PHF1 are significantly reduced in both soluble and insoluble fractions (Student's $t$-test, ***$P = 0.00072$ and **$P = 0.004$, for B; and **$P = 0.007$ and **$P = 0.009$ for C).

D    Western blot analysis of total Tau, unphosphorylated Tau (Tau1) and CP13- or PHF1-positive pTau levels in response to doxycycline induction (Tau) and/or TFEB-FLAG transfection (TFEB) in T40PL cell line. Ctrl: untreated; Tau: 0.5 µg/ml doxycycline treated for 48 h; TFEB: TFEB transfected for 48 h; Tau + TFEB: combined TFEB transfection and DOX treatment. FLAG: anti-FLAG antibody blotting for TFEB expression. γ-tubulin was used as a loading control.

E    Quantification of relative band intensities. CP13, PHF1 and total Tau are significantly reduced by TFEB transfection (Student's $t$-test, $n = 3$, **$P = 0.0012, 0.0040$, and $0.0019$, respectively). Each bar represents average $\pm$ s.e.m.

Source data are available online for this figure.

staining in GFP/TFEB coinjected mice (Fig 3A), suggesting that TFEB mediates pTau clearance in a cell autonomous manner. The same result was also obtained when AAV-GFP/TFEB was coinjected into 2-month-old Tau mice (Supplementary Fig S4). To corroborate the *in vivo* findings, we transfected EGFP or TFEB-FLAG expression vectors in the doxycycline-induced T40PL cell line and performed double immunofluorescence staining of GFP or TFEB with PHF1 every 8 h up to 40 h (Fig 3B). The results show that TFEB intensities begin to negatively correlate with PHF1 staining staring at the 16 h time point and persist to 24, 32 (not shown) and 40 h, while GFP intensities show no correlation with PHF1 at all time points examined (Fig 3B and quantified in Fig 3C). Furthermore, consistent with the *in vivo* results, while transfection of the GFP vector revealed abundant PHF1- and GFP-double-positive cells, most of the TFEB-positive cells exhibited reduced PHF1 intensities (insets in Fig 3B). These results provide strong support that TFEB mediates time-dependent pTau protein clearance through a cell autonomous mechanism.

**TFEB rescues neurodegeneration in rTg4510 mice**

In agreement with the published reports, immunostaining using the neuronal marker NeuN reveal severe neurodegeneration in rTg4510 Tau mice (Fig 4A and B, WT vs. Tau). Life-long expression of TFEB did not lead to detectable alteration of neuronal structure in wild-type mice (WT vs. WT + TFEB), but resulted in grossly expanded hippocampal sizes in Tau transgenic background (Tau vs. Tau + TFEB). Enhanced neuronal survival is confirmed by quantifying neuronal numbers in the CA1 area of hippocampus using unbiased stereology (Fig 4C). In fact, a beneficial effect of TFEB is readily appreciable by measuring the brain weight, which documented the same brain weight in wild-type mice regardless of TFEB expression (Fig 4D, WT vs. WT + TFEB), but a significant increase in TFEB-treated Tau mice compared to uninjected Tau controls (Tau vs. Tau + TFEB). Improved overall neuronal health is evidenced by reduced neuroinflammation detected by Iba1 (Fig 4E) and GFAP (Fig 4F).

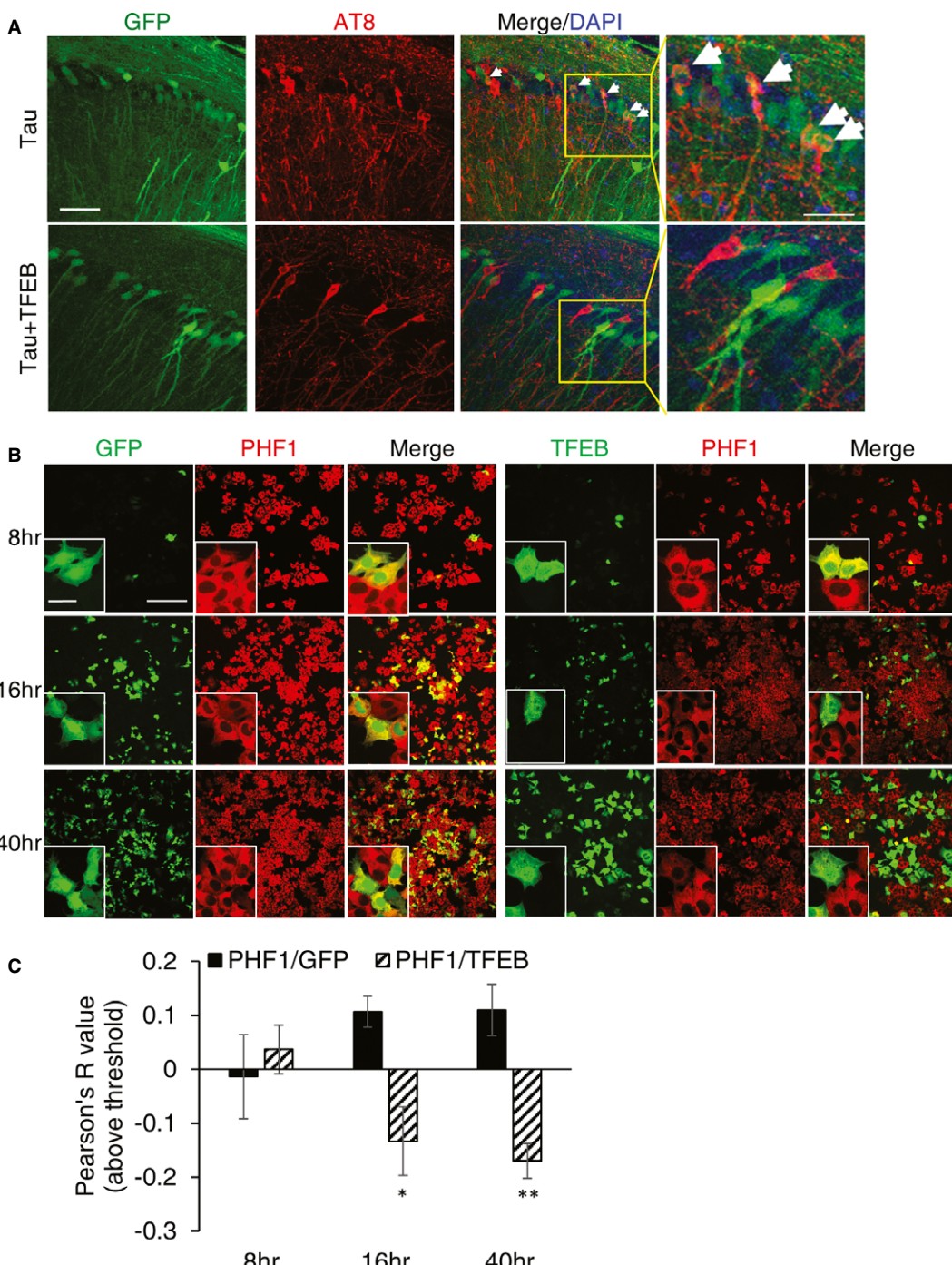

**Figure 3.  Intracellular clearance of pTau pathology by TFEB.**

A   Immunohistochemical staining of hippocampus of rTg4510 (Tau) mice injected with AAV-GFP (Tau) or AAV-TFEB/AAV-GFP (Tau + TFEB) at P0 and analyzed at 4 months. Images are displayed as GFP only (GFP), AT8 only (AT8) or overlay of GFP/AT8/DAPI (Merge/DAPI). Right panels are enlarged images of the bracketed areas. Arrowheads indicate GFP/AT8-double-positive cells only in GFP-injected mice. Scale bar: 50 μm. $n = 2$ for AAV-GFP and $n = 5$ for AAV-TFEB/AAV-GFP; 4–5 sections/ mouse were examined.

B   Fluorescence images of GFP and immunofluorescence images of anti-FLAG compared with immunofluorescence images of anti-PHF1 in GFP or TFEB transfected and doxycycline induced T40PL cell line at 8, 16 or 40 h post-transfection. Merge: Overlay of GFP or TFEB-FLAG with PHF1. Insets are higher resolution images documenting that whereas overlapping GFP and PHF1 immunoreactivity can be observed at all the time points, TFEB- and PHF1-double-positive cells can only be detected at 8 h but not later times. Scale bar: 200 μm; in inset: 50 μM.

C   Pearson correlation coefficients (Pearson R-values) showing no correlation between GFP and PHF1 (PHF1/GFP) at any time points, but negative correlation between TFEB and PHF1 (PHF1/TFEB) starting at 16 h and persisting to 40 h (Student's *t*-test, $n = 4$, *$P = 0.026$ and **$P = 0.008$ for 16 and 40 h, respectively, comparing R-values for GFP/PHF1 with TFEB/PHF1). The whole view field of eight confocal projection slices per view field, four view fields per time point per transfection was used for analysis. Bar graph represents average ± s.e.m.

**Figure 4.   TFEB ameliorates neuronal loss and neuroinflammation.**

A    Immunofluorescence staining of untreated or TFEB-treated (+TFEB) hippocampus of wild-type (WT) and rTg4510 (Tau) mice using the anti-NeuN antibody. Scale bar: 1000 μm.

B    Enlarged view of the bracketed areas in A. Scale bar: 50 μm.

C    Unbiased stereological quantification of NeuN-positive neurons in area CA1 of wild-type (WT) or Tau transgenic mice injected with GFP or TFEB. $n$ = 4 sections/ mice, 5 mice/group. Student's $t$-test, **$P$ = 0.0063.

D    Wet brain weight measurement of WT or Tau mice with (+TFEB) or without TFEB injection. $n$ = 14 mice/group. ***$P$ < 0.001 (2 way ANOVA with Bonferroni post hoc). Each bar represents average ± s.e.m.

E, F    Representative immunofluorescence images of cortex (CTX) or hippocampus (HPC) of untreated or TFEB-treated (+TFEB) rTg4510 (Tau) mice stained with anti-Iba1 (E) or anti-GFAP (F) antibodies antibody. Scale bar in E: 200 μm; in F: 400 μm.

## TFEB improves cognitive performance and synaptic function

To investigate whether TFEB-mediated biochemical and morphological changes in rTg4510 mice were accompanied by improved functional outcome, we performed Morris water maze (MWM) test in 4-month-old rTg4510 Tau mice and wild-type littermates with or without TFEB treatment (Fig 5A and B). P0-injected mice were used since widespread brain expression can be achieved by this approach. Mice were trained with blocks of 4 trials at 2 blocks per day over 4 days, followed by memory testing in a probe trial. Wild-type mice with or without TFEB treatment displayed similar latency during the training phase and platform crossing in the probe trial

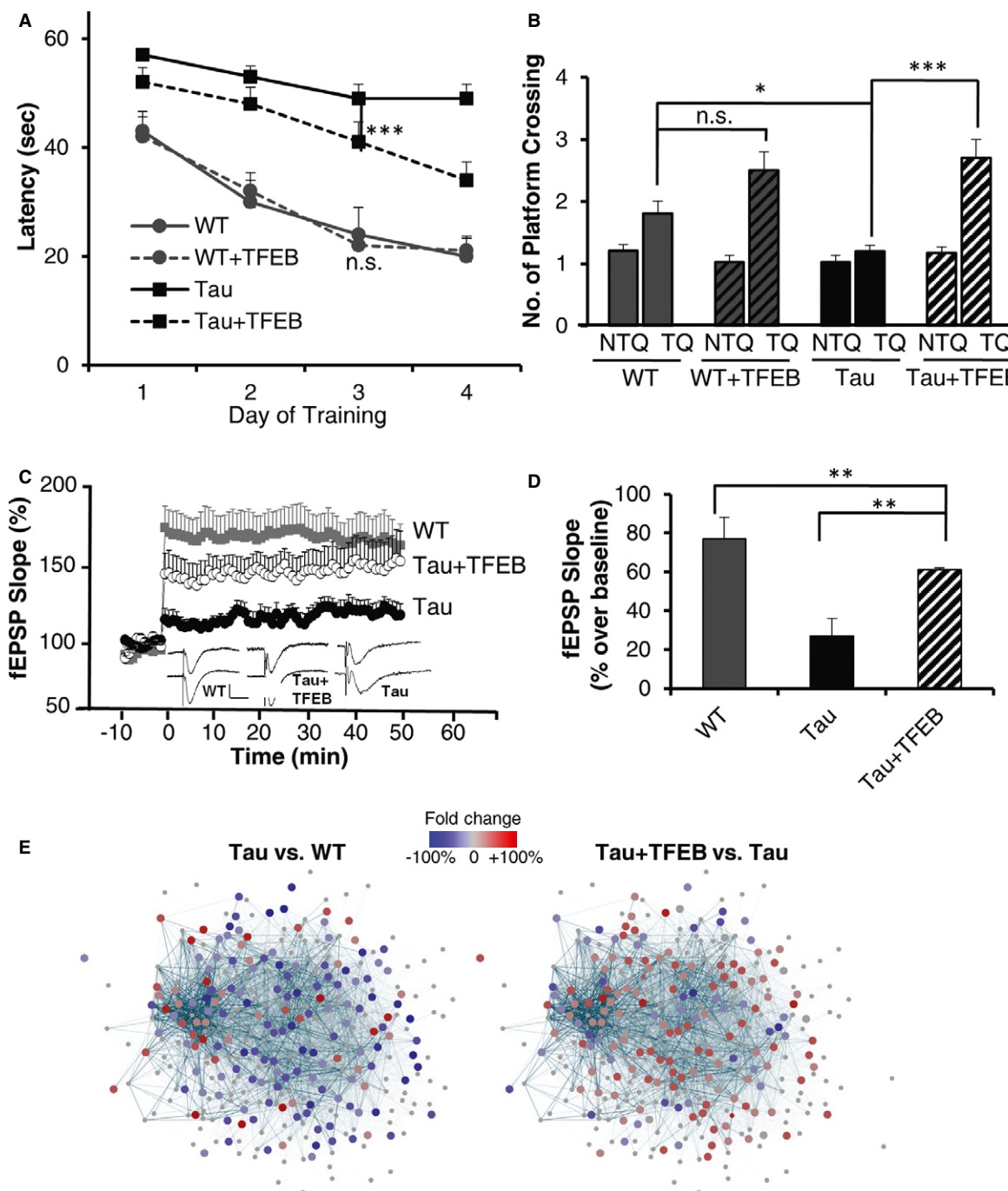

**Figure 5.   Life-long TFEB treatment improves behavioral performance and synaptic function.**

A   Morris water maze test of groups of 4-month-old WT or Tau mice without or with TFEB (+TFEB) treatment, showing longer latency (time to find hidden platform) in Tau mice compared to WT controls at any given day of training. TFEB has no effect on WT mice, but significantly reduced the latency in Tau mice. *n* = 14/group. ***$P$ < 0.001 (2 way ANOVA).

B   Number of platform crossings in non-target quadrants (NTQ) vs. the target quadrant (TQ). Values of the three NTQs were combined and averaged. n.s.: non-significant. *$P$ = 0.047 and ***$P$ = 0.0005 for TQ of WT vs Tau and Tau vs Tau + TFEB, respectively (Student's *t*-test). Each bar represents average ± s.e.m.

C   Slope of field excitatory postsynaptic potential (fEPSP) in response to theta-burst stimulation delivered to the Schaffer collateral pathway from WT mice (*n* = 10 recordings from 6 mice), Tau mice (*n* = 19 recordings from 6 mice) or Tau mice with TFEB injection (Tau + TFEB; *n* = 8 recordings from 6 mice). Insets: example fEPSP traces taken before (upper) or after (lower) stimulation from WT, Tau or Tau + TFEB mice. Calibration: 1 mV, 5 msec.

D   Quantification of average fEPSP slope in the last 10 minutes demonstrating severely impaired LTP in Tau mice (WT vs. Tau) and significant improvement by TFEB treatment (Tau vs. Tau + TFEB). Each bar represents average ± s.e.m. **$P$ < 0.01 (2 way ANOVA with Bonferroni post hoc).

E   Cytoscape-generated networks representing genes involved in synaptic function. A majority of downregulated genes (blue dots) are found in the synaptic network when comparing transcription profiles of Tau mice with wild-type controls (Tau vs. WT), while a majority of upregulated genes (red dots) are found when comparing TFEB-injected with uninjected Tau mice (Tau + TFEB vs. Tau). *n* = 4 mice/genotype/treatment group.

(WT vs. WT + TFEB), indicating that TFEB expression caused no adverse effect in learning and memory in wild-type mice. rTg4510 mice performed poorly in both the latency and probe tests. Expression of TFEB significantly improved learning and enhanced memory retention as evidenced by shorter latency with training and higher platform crossing compared to Tau mice without TFEB injection (Tau vs. Tau + TFEB).

Synaptic plasticity has long been proposed to be a cellular mechanism underlying learning and memory. Given that we observed a significant improvement in the hippocampal-dependent behavior with TFEB, we recorded field Schaffer collateral long-term potentiation (LTP) in acute hippocampal slices of Tau mice with or without TFEB treatment (Fig 5C). Because the uninjected and injected wild-type mice exhibited similar behavioral performance, we collected parallel recordings only on the uninjected group. Compared to wild-type controls, the slope of field excitatory postsynaptic potentials (fEPSPs) induced by theta-burst stimulation was greatly reduced and remained low in Tau mice (Fig 5C, WT vs. Tau). Expression of TFEB partially but significantly rescued the LTP defects (Fig 5D, Tau vs. Tau + TFEB). The improvement of neuronal function by TFEB is supported by expression microarray analysis of hippocampal samples. In agreement with the published report (Kopeikina et al, 2012), we found downregulation of many of the synaptic protein genes in Tau mice compared to wild-type controls when mapped on a synaptic gene network obtained by pathway co-expression analysis (Tau vs. WT, Fig 5E and Supplementary Fig S5) (Palmieri et al, 2011; The Gene Ontology Consortium, 2012; Song et al, 2013). These downregulated genes were largely restored upon TFEB expression (Tau + TFEB vs. Tau, Fig 5E and Supplementary Fig S5). The fact that we did not find significant changes of the synaptic pathway genes by comparing wild-type mice with or without TFEB injection (not shown) supports the notion that the increased synaptic gene expression in TFEB-treated Tau mice is due to the rescue of Tau-triggered synaptic protein reduction rather than a dominant effect of TFEB.

### Activation of autophagy and lysosomal pathways by TFEB

Because TFEB has been shown to regulate the ALP through direct activation of autophagy and lysosomal target genes (Sardiello et al, 2009; Settembre et al, 2011), we performed Gene Set Enrichment Analysis (GSEA) (Subramanian et al, 2005) of transcriptome changes of lysosomal and autophagy genes in TFEB-injected vs. uninjected hippocampal samples. As expected, we found global enrichment of lysosomal target genes (Fig 6A). Consistent with the transcriptional upregulation of lysosomal genes, immunostaining of LAMP1 in AAV-TFEB-injected brains documents that nuclear TFEB expression is associated with higher LAMP1 levels (Fig 6B). This is also corroborated by immunoblotting of LAMP1 and another lysosomal enzyme, CTSD, which reveal that TFEB injection is associated with mild but significant increases of these lysosomal protein levels (Fig 6C and quantified in 6D).

However and unexpectedly, we did not detect appreciable transcriptional upregulation of autophagy pathway molecules as a function of TFEB expression (Supplementary Fig S6). qRT-PCR analysis of selected TFEB lysosomal and autophagy targets confirmed the microarray data (Supplementary Fig S6B and C).

Since ALP can be regulated at both transcriptional and post-transcriptional levels, we examined whether the autophagy pathway can be activated by TFEB in general. We assessed the steady-state levels of LC3-II, the best characterized marker of the autophagosome, in TFEB-transfected T40PL cells. As expected, TFEB overexpression increased autophagosome formation, as demonstrated by immunoblot analysis showing increased levels of LC3-II (Fig 7A). However, this activity is sensitive to the duration of TFEB transfection. Specifically, a prominent induction of LC3-II can only be detected 24–48 h post-transfection. LC3-II levels returned to baseline afterward (Fig 7A). Downregulation of autophagy at later time points (72 h) is accompanied by increased LAMP1 levels, indicating that TFEB could dynamically regulate ALP by coordinating autophagy activation with lysosomal clearance.

To provide evidence that TFEB-mediated ALP is involved in pTau clearance, we cotransfected T40PL cells with TFEB and a monomeric RFP-GFP-tagged LC3 construct, which serves as a reporter for the autophagy flux (Kimura et al, 2007). As expected, both RFP-only autolysosomes (red arrows) and GFP/RFP double-positive autophagosomes (yellow arrow) can be detected (Fig 7B upper inset). Compared to vector transfected controls, there was an increase in the number of autolysosomes as a function of TFEB expression (Supplementary Fig S7). Immunostaining with PHF1 showed that PHF1-positive staining can be detected within autophagosomes recognized by the double membrane LC3 puncta (Fig 6B inset on right) and that cells with significant LC3 puncta (Fig 7B, thick white arrows), indicating increased autophagy, had dramatically lower PHF1 levels compared to nearby LC3 puncta-negative cells. These results suggest that pTau is recruited to the autophagosomes and that autophagy activation is associated with pTau degradation. However, transfection of T40PL cells with lyso-tracker-red followed by staining with LC3 and PHF1 revealed the presence of PHF1 in both LC3 puncta-positive and -negative lysosomes, indicating both autophagy-dependent and independent lysosomal degradation of pTau (Supplementary Fig S8).

A functional role of TFEB-mediated lysosomal activation in pTau clearance is evidenced by coimmunostaining of LAMP1 and pTau in TFEB-transfected cells, which reveal that TFEB nuclear staining is associated with higher LAMP1 and lower PHF1- and MC1-positive Tau (Fig 7C). Direct evidence that pTau is processed through ALP is documented by showing that treatment of Tau cells with Leupeptin, a lysosomal cysteine and serine protease inhibitor, resulted in increased total Tau and PHF1 Tau under both basal and TFEB-transfected conditions (Fig 7D and E). Thus, the combined results implicate an important role of the lysosome in TFEB-mediated pTau clearance.

### PTEN is a direct target of TFEB that mediates pTau clearance

Although we failed to detect global autophagy gene induction by TFEB, analysis of the microarray data followed by qRT-PCR revealed that expression of phosphatase and tensin homolog (PTEN), a lipid phosphatase that antagonizes the phosphatidylinositol-3-kinase (PI3K)-Akt-mTOR signaling (Kwon et al, 2001, 2003), was significantly elevated in TFEB-treated WT and Tau mice (Fig 8A). Interestingly, examination of the PTEN promoter identified 2 putative CLEAR sequences (Fig 8B), indicating that TFEB may activate PTEN by binding to the CLEAR motifs. We thus carried out chromatin

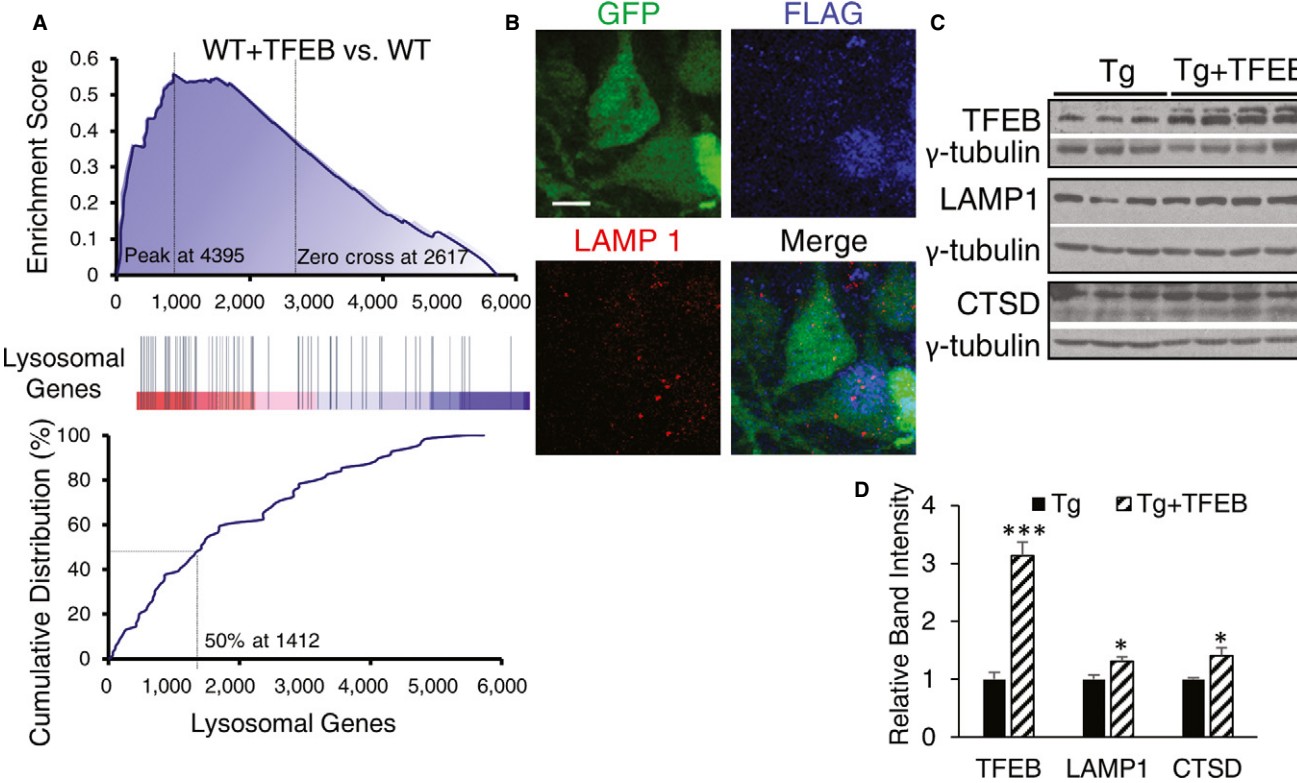

**Figure 6.  Analysis of TFEB-mediated lysosomal gene activation *in vivo*.**

A    Gene set enrichment analysis (GSEA) of transcriptome changes in TFEB-injected vs. uninjected wild-type (WT + TFEB vs. WT) mice. GSEA of genes annotated as participating in the lysosomal function are reported. The upper panel shows the enrichment generated by GSEA of ranked gene expression data (left in upper panel and red in middle panel: upregulated; right in upper panel and blue in middle panel: downregulated). Peak value corresponds to the point (along the ranked microarray) where the enrichment score (ES) meets its highest value, while the zero value indicates the boundary (along the ranked microarray) between positive fold change and negative fold change. In the middle panel, vertical blue bars indicate the position of lysosomal genes within the ranked list. Lower panel shows the cumulative distribution of lysosomal genes within the ranked lists. The ranking positions that include 50% of lysosomal genes are indicated. The analysis shows that lysosomal genes have a significant global shift toward upregulated genes in TFEB-injected mice compared with uninjected littermates (ES = 0.56, $P$ = 0.0029). $n$ = 4 mice/group.

B    Representative image of Tau mice coinjected with TFEB-FLAG/GFP and immunostained with anti-FLAG and LAMP1 antibodies. Merge image highlights that nuclear TFEB-FLAG is correlated with higher LAMP1. Scale bar: 10 μm.

C    Western blot analysis of TFEB, LAMP1 and CTSD expression in Tau mice with (+TFEB) or without AAV-TFEB adult injection.

D    Quantification of relative band intensities of (C). $n$ = 3 and 4 per group. TFEB protein levels are significantly increased in Tg + TFEB vs Tg (***$P$ = 0.00013) along with LAMP1 protein levels (*$P$ = 0.013) and CTSD protein levels (*$P$ = 0.028), Student's $t$-test. Each bar represents average ± s.e.m.

Source data are available online for this figure.

immunoprecipitation (ChIP) experiment using HeLa cells stably expressing the TFEB-FLAG vector (Sardiello *et al*, 2009). Anti-FLAG antibody pull-down followed by PCR amplification showed that, indeed, a CLEAR-containing PTEN promoter fragment was enriched in the anti-TFEB-FLAG immunoprecipitants (Fig 8C). To establish a functional role of the TFEB/CLEAR interaction, we performed luciferase reporter assay by cotransfecting a TFEB expression vector with PTEN-luciferase reporters with or without CLEAR motifs in N2a cells (Teresi *et al*, 2008). We observed significant activation of luciferase activities only in constructs containing one or both CLEAR sequences (−1334 to +1 and −453 to +1), but not in the vector without the CLEAR motif (−203 to +1) (Fig 8D). The increase in luciferase activity is TFEB dose-dependent (Fig 8E). Due to PTEN's important roles in regulating various cellular mechanisms, its expression at both mRNA and protein levels are controlled by intricate regulatory mechanisms ((Song *et al*, 2012) for a recent review).

We only observed a 19% increase in PTEN protein level upon AAV-TFEB P0 injection in wild-type mice (Supplementary Fig S9A and B). A small (14%) but statistically significant protein level increase is also observed in adult injected mice (Supplementary Fig S9C and D). The relatively small increase of PTEN protein levels resulting from p0 and adult mice TFEB injection make it difficult to ascertain whether these changes drive pTau clearance. In order to further validate the connection between TFEB and PTEN, we took advantage of the *in vitro* system where TFEB can be transiently overexpressed more than 20-fold and assayed PTEN expression thereafter. Western blot analysis showed that transfection of TFEB directly drives endogenous PTEN protein expression, and is associated with elevated LC3-II levels (Fig 8F and G). Immunostaining for TFEB, PTEN and pTau reveals that TFEB overexpression is correlated with increased PTEN and reduced PHF1-positive Tau (Fig 8H). As expected, enhanced TFEB-dependent PTEN upregulation is associated

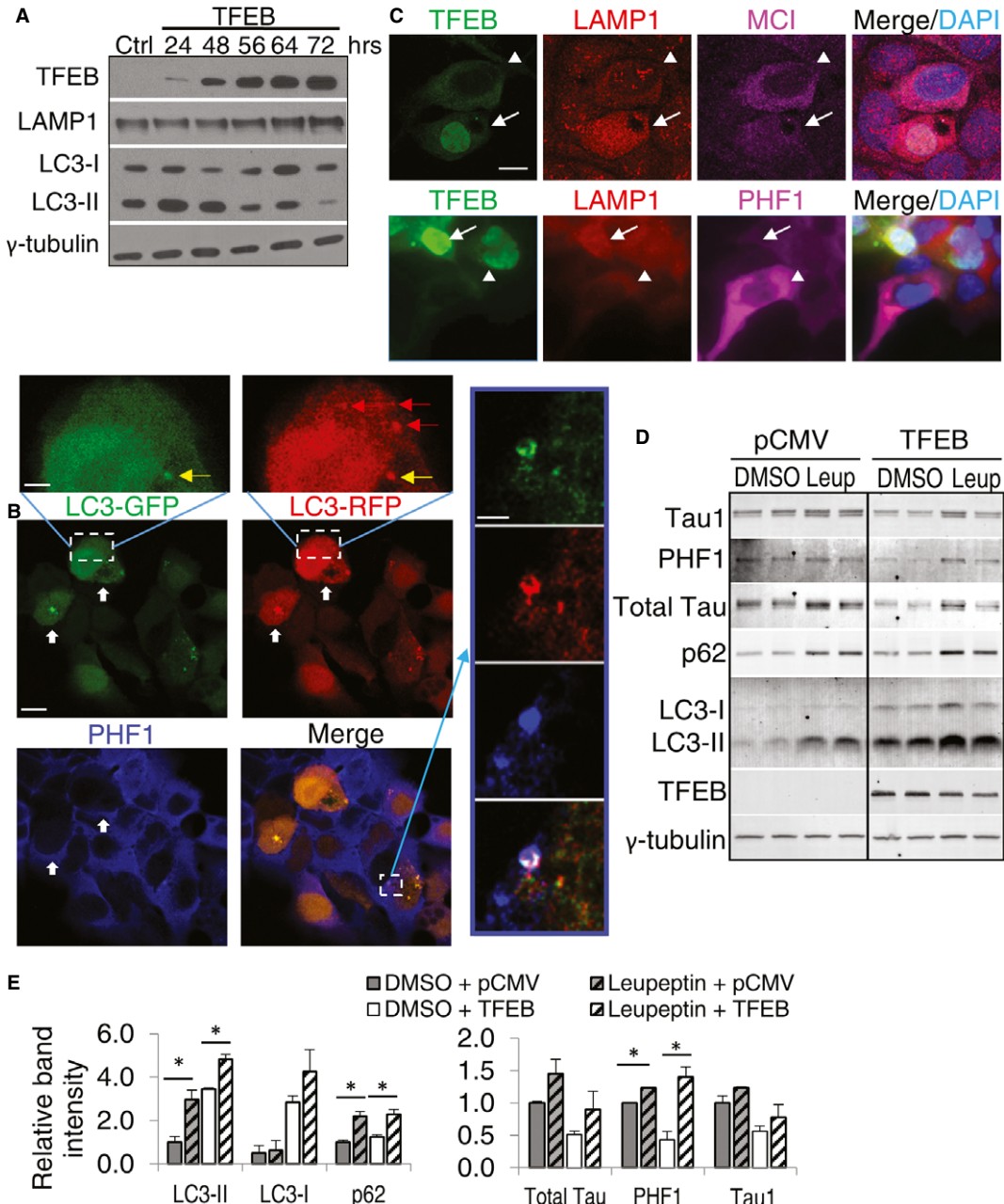

**Figure 7.  Analysis of TFEB-mediated autophagy and lysosomal pathway *in vitro*.**

A   Western blot analysis of Lamp1 and LC3-I and LC3-II levels in T40PL cells transfected with TFEB-FLAG and harvested at times indicated. γ-tubulin was used as a loading control. Ctrl: untransfected.

B   Fluorescent images of T40PL cells transfected with TFEB and RFP-GFP-LC3. Thick white arrows indicate cells with significant LC3 puncta and lower PHF1 compared with nearby cells. Images on the upper and right are higher magnification views of the bracketed area of the LC3-GFP and LC3-RFP or Merge panels, respectively. RFP-only autolysosomes are marked by red arrows; GFP/RFP double-positive puncta representing an autophagosome is shown in yellow (upper inset). Right insets highlight a cell with colocalization of GFP/RFP-positive autophagosome with PHF1. Scale bar: 10 μm; in inset: 2.5 μm.

C   Immunofluorescence images of T40PL cells transfected with TFEB and stained with anti-LAMP1/MC1 (top) or PHF1 (bottom). Arrow marks cells with nuclear TFEB and its correlation with higher LAMP1 and lower MC1 (top) or PHF1 (bottom) stainings. Arrowhead indicates nearby cells with cytoplasmic TFEB and opposite LAMP1 and MC1/PHF1 patterns. Scale bar: 10 μm.

D   Western blot analysis of T40PL cells transfected with either pCMV or TFEB, allowed to recover for 24 h, and then treated with 0.5 μg/ml doxycycline and either DMSO or 50-μM Leupeptin for 48 h prior to lysis. Black line denotes cropped lanes from a single immunoblot.

E   Quantization of the autophagy markers LC3-I, LC3-II (pCMV vs pCMV+Leupeptin $P = 0.032$, TFEB vs TFEB+Leupeptin $P = 0.014$), and p62 (pCMV vs pCMV+Leupeptin $P = 0.019$, TFEB vs TFEB+Leupeptin $P = 0.024$), and Tau species markers Tau1, PHF1 (pCMV vs pCMV+Leupeptin $P = 0.00082$, TFEB vs TFEB+Leupeptin $P = 0.022$), and total Tau normalized for loading to γ-tubulin from (D). *P*-values determined by *t*-test, $n = 4$. (*$P < 0.05$) Each bar represents average ± s.e.m.

Source data are available online for this figure.

**Figure 8.   PTEN is a direct target of TFEB.**

A   qRT-PCR analysis of PTEN levels as a function of TFEB treatment in WT or Tau mice. n = 3 mice/genotype/treatment group in triplicates. *P = 0.027 and **P = 0.01 for WT vs WT + TFEB and Tau vs Tau + TFEB, respectively (Student's t-test).

B   Putative CLEAR sequences in the PTEN promoter and their positions. TSS: translation start site. −203 to +1, −453 to +1 and −1334 to +1: PTEN promoter fragments linked to the luciferase reporter.

C   ChIP analysis of TFEB binding. HPRT and APRT: negative controls; MCOLN1: positive control. Open bar: IgG control; filled bar: TFEB-FLAG IP. The experiment was done two times each in triplicates.

D   Normalized luciferase activity by transfecting 300 ng of each PTEN-luciferase reporter. Open bar: vector transfection control; filled bar: TFEB transfected. Student's t-test, n = 4; *P = 0.01; **P = 0.005

E   Dose-dependent luciferase activities in response to increasing concentrations of TFEB using the (−1344 to +1) PTEN-luciferase reporter. The experiment was done three times each in triplicates.

F   Western blot analysis of PTEN and LC3 expression in T40PL cells transfected with either pCMV or TFEB.

G   Quantization of blots in F and normalized for loading to γ-tubulin. PTEN and LC3-II are significantly increased in cells transfected with TFEB with *P = 0.032 and 0.036, respectively (Student's t-test, n = 3).

H   Representative immunofluorescent images of T40PL cells transfected with TFEB and triple staining for TFEB (FLAG), PTEN and PHF1. Arrow marks cell with higher TFEB and PTEN is correlated with lower PHF1. Arrowhead indicates nearby cell with opposite TFEB/PTEN/PHF1 patterns. Scale bar: 10 μm. Each bar represents average ± s.e.m.

Source data are available online for this figure.

with reduced pAkt (S473), pP70SK6 (T389) and pULK1 (S757), but not their total protein levels (Supplementary Fig S10).

Previous reports have implicated a role of PTEN in pTau reduction (Kerr *et al*, 2006; Zhang *et al*, 2006a,b). To determine whether PTEN expression leads to reduced pTau levels in our system, we transfected either an empty vector (Ctrl), a wild-type PTEN expression vector (WT) or the phosphatase (C124S) or lipid phosphatase (G129E) deficient mutants in T40PL cells and measured total and phosphorylated Tau levels upon doxycycline induction (Wang *et al*, 2007). Biochemical (Fig 9A and B) and immunohistochemical (Fig 9C) analyses showed that only the wild-type PTEN and not the phosphatase-defective mutants were associated with reduced PHF1 Tau, demonstrating a role for PTEN in pTau reduction that requires its lipid phosphatase activity. To ascertain whether PTEN is required in mediating TFEB-dependent pTau reduction, we cotransfected either an EGFP or TFEB expressing vector with a PTEN shRNA construct or a scrambled control into Tau expressing cells (Fig 9D). Expression of the PTEN shRNA markedly blunted TFEB-mediated PHF1-positive pTau and total Tau reduction and LC3-II elevation (Fig 9E), demonstrating that PTEN is critical in mediating TFEB-dependent pTau clearance.

## Discussion

In this report, we investigated the role of TFEB on AD neuropathology by directly injecting AAV-TFEB into 5xFAD and rTg4510 transgenic mouse brains, which affords widespread and persistent TFEB expression *in vivo*. We demonstrate that TFEB exerts a potent activity in attenuating NFT pathology without overtly affecting Aβ deposition or displaying adverse reactions in wild-type mice. The reduction of pTau/NFTs by TFEB is accompanied by improved neuronal survival and function. In light of the substantial evidence supporting a role of ALP in APP turnover and Aβ catabolism (Mueller-Steiner *et al*, 2006; Nixon, 2007; Pickford *et al*, 2008; Rohn *et al*, 2011; Yang *et al*, 2011), our finding that TFEB has no appreciable effect on Aβ pathology is somewhat unexpected. It is important to note, however, this negative data should be interpreted with caution. Our finding is limited to the system we employed using the 5xFAD mice and needs to be further validated in other APP/Aβ mouse models and by other approaches such as genetic manipulation. It remains possible that the TFEB expression attainable in our system may not be sufficient to impact Aβ pathology or that APP/Aβ may be subject to TFEB independent ALP regulation. Nevertheless, the potent clearance of pTau/NFTs using the same TFEB injection scheme argues that Aβ and pTau/NFT pathologies are subject to distinct TFEB regulations.

Biochemical analysis of rTg4510 mouse brain samples and T40PL cell lysates revealed that TFEB enhances clearance of only the hyperphosphorylated and misfolded Tau proteins. The fact that the Tau1-positive unphosphorylated Tau remained constant in all preparations strongly supports the notion that TFEB leads to degradation rather than dephosphorylation of the aberrant Tau species. Nevertheless, although it is clear that TFEB targets the detergent-insoluble pTau for degradation, we cannot exclude the possibility that TFEB may promote pTau dephosphorylation or lower the rate of Tau phosphorylation in the soluble pool. Since TFEB recognizes Tau phosphorylated at multiple sites as well as misfolded Tau marked by MC1 immunoreactivity, we favor a mechanism in which

TFEB primarily triggers the degradation of the misfolded Tau species including but not limited to hyperphosphorylated Tau. However, it is also possible that TFEB-mediated clearance recognizes only the Tau species marked by hyperphosphorylation and that the clearance of MC1-positive species is the result of hyperphosphorylation. Our results that TFEB and pTau stainings are mutually exclusive support a neuronal-intrinsic role of TFEB in pTau reduction. However, it remains possible that other cell types, such as astrocytes and microglia, may also participate in TFEB-dependent pTau clearance in a non-cell-autonomous manner. In this regard, it is interesting to note that TFEB has been reported to mediate not only lysosomal biogenesis but also lysosomal exocytosis (Medina *et al*, 2011). It is difficult to probe the cellular mechanisms using the current system as the AAV vector infects both neurons and non-neuronal cells. Genetic expression of TFEB in these distinct cell types is better suited to address this question.

TFEB has been reported to regulate autophagy through transcriptional activation of autophagy genes (Settembre *et al*, 2011). Our gene expression analysis failed to detect significant upregulation of autophagy targets in TFEB-treated samples. This could be attributed to the modest TFEB expression or the tissue-specific regulation of autophagy gene induction by TFEB. Regardless, our cell culture studies support a role of TFEB in the dynamic regulation of ALP by coordinating autophagy activation with lysosomal clearance and our analysis implicates this regulation, in particular lysosomal activity, in TFEB-mediated pTau clearance. Importantly, we identify PTEN as a *bona fide* target of TFEB that may mediate the TFEB effect on pTau reduction. Since PTEN could activate autophagy by antagonizing the PI3K-Akt-mTOR signaling (Kwon *et al*, 2001, 2003), it is reasonable to hypothesize that the TFEB-PTEN-PI3K-Akt-mTOR signaling pathway is involved in the pTau clearance through ALP. This notion, combined with the published reports showing that TFEB is a substrate of mTOR (Roczniak-Ferguson *et al*, 2012; Settembre *et al*, 2012), raises an interesting possibility for a TFEB-PTEN-Akt-mTOR-TFEB feedback regulatory loop whereby TFEB induces autophagy through upregulation of PTEN and inhibition of Akt and mTOR, which in turn results in further TFEB activation. Nevertheless, enhanced lysosomal activity by TFEB may also play crucial roles in pTau/NFT degradation by effectively clearing LC3II-positive autophagosomes and/or degrading pTau independent of macroautophagy. Further studies are needed to delineate the contribution of autophagy-dependent vs. autophagy-independent lysosomal clearance of pTau and to what extent PTEN-mediates ALP downstream of TFEB. In addition, it is important to point out that *in vitro* cell cultures were used for mechanistic studies. The nature of pTau expressed in the acutely induced cells versus *in vivo* is likely distinct, especially with regards to its aggregation status. Therefore, the relevance of the signaling pathways identified here to tauopathies *in vivo* requires further investigation and validation.

The mechanism mediating the macroautophagy-independent lysosomal degradation of pTau remains to be investigated. One possibility is that clearance occurs through chaperone-mediated autophagy (CMA). In this regard, Tau has been reported to contain two CMA targeting motifs obligatory for hsc70 binding and LAMP2A-mediated lysosomal degradation (Wang *et al*, 2009). However, only proteolytically processed Tau fragments were shown to be subject to the CMA pathway (Wang *et al*, 2009). Since we did not detect appreciable levels of Tau fragments, the role of CMA in

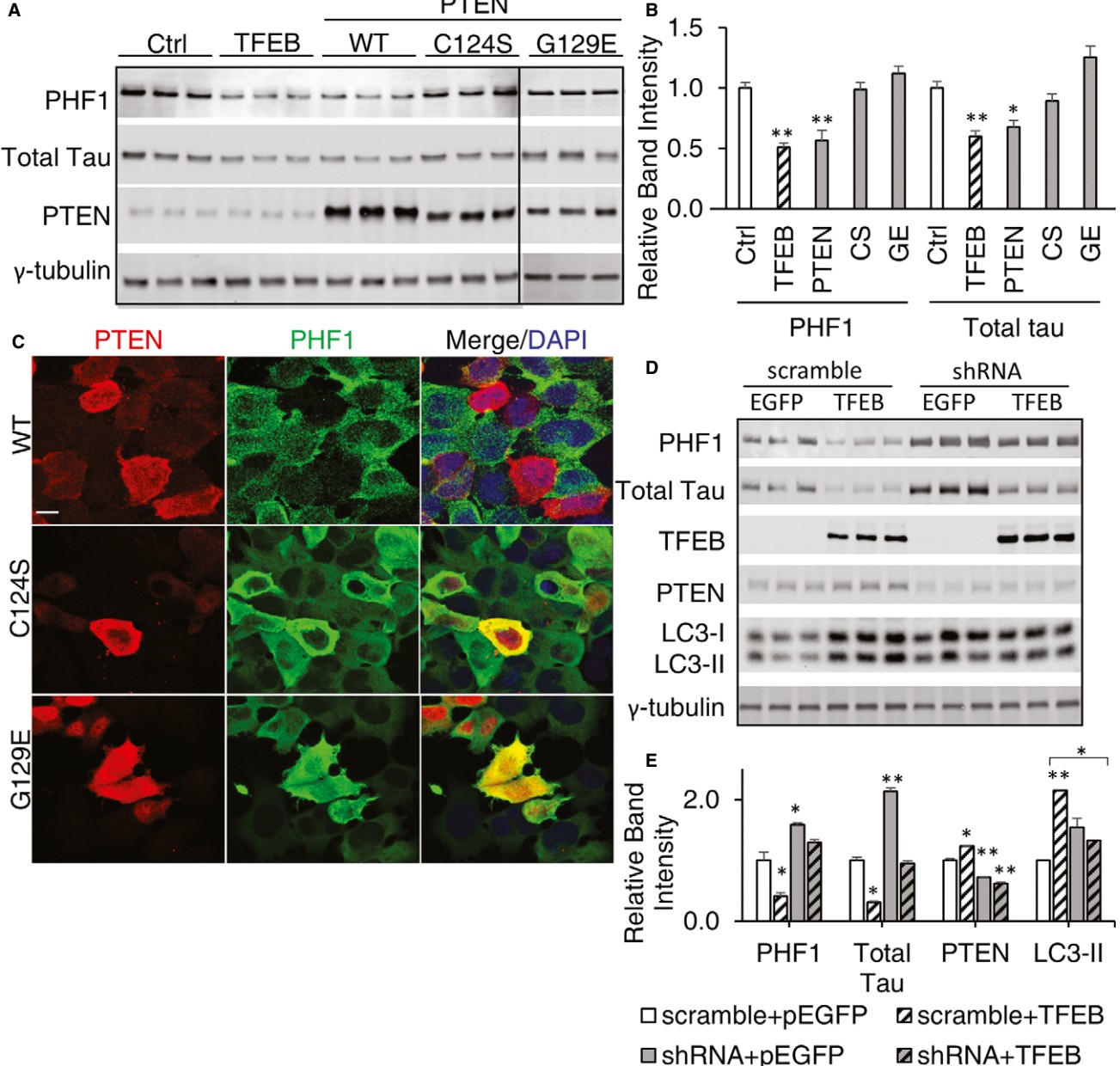

**Figure 9.  TFEB mediates pTau reduction in a PTEN-dependent manner.**

A   Western blot analysis of total Tau and PHF1-positive pTau levels as a function of wild-type or the C124S or G129E mutant PTEN expression. Empty vector (Ctrl)- or TFEB-transfected cells were used as negative and positive controls, respectively. γ-tubulin was used as a loading control.

B   Quantization of blots in A and normalized for loading to γ-tubulin. TFEB and PTEN transfection significantly reduced pTau as probed by PHF1 (*P* = 0.0095 and 0.0092, respectively), as well as total Tau (*P* = 0.0052 and 0.034, respectively), *n* = 3, Student's *t*-test.

C   Representative immunofluorescence imaging showing that cells with wild-type PTEN (WT), but not the C124S or G129E mutants, were correlated with reduced PHF1-positive Tau. Scale bar: 10 μm.

D   Western blot analysis of total Tau, PHF1-positive pTau and LC3 levels in response to TFEB or TFEB and PTEN shRNA expression. Transfection of a scramble PTEN shRNA was used as negative controls. γ-tubulin was used as a loading control.

E   Quantization of blots in D and normalized for loading to γ-tubulin. Comparing with cells transfected with scramble+pEGFP, scramble+TFEB transfection reduces PHF1 and total Tau intensity (*P* = 0.038 and 0.035, respectively) and increases PTEN and LC3II (*P* = 0.04 and 0.0076, respectively) (white slashed bars). shRNA significantly reduce PTEN when transfected with either pEGFP or TFEB (*P* = 0.0089 and 0.043, respectively), but TFEB's effect in reducing pTau drops from 60% reduction in the presence of scramble sequence to 30% reduction when combined with shRNA of PTEN. All statistics here are Student's *t*-test, *n* = 3. Each bar represents average ± s.e.m.

Data information: \**P* < 0.05, \*\**P* < 0.01, exact value described above for each analysis.
Source data are available online for this figure.

our system is not clear. Besides autophagy, PTEN may mediate Tau homeostasis through additional activities. In particular, the downregulation of pAkt by PTEN may directly affect Tau phosphorylation status independent of mTOR (Zhang *et al*, 2006a,b). Akt may also participate in ubiquitin-proteasome degradation of pTau species by interacting with the HSP90/CHIP ubiquitin ligase complex (Dickey *et al*, 2007, 2008). Further, since PTEN has been shown to be essential for neural development and synaptic plasticity (Kwon *et al*, 2006; Zhou *et al*, 2009; Sperow *et al*, 2011; Takeuchi *et al*, 2013), upregulation of PTEN may directly contribute to the improved neuronal survival and synaptic function in TFEB-treated Tau mice.

Our studies implicate mTOR as an integral component of the TFEB-PTEN-Akt-mTOR autophagy pathway, which is compatible with a recent report documenting that TFEB rescued α-synuclein toxicity through downregulation of mTOR (Decressac *et al*, 2013). However, a potent role of TFEB in lysosomal biogenesis distinguishes the effects of TFEB from autophagy activators mechanistically and functionally. Of particular significance, while autophagy leads to decreased endogenous Tau and degradation of pTau/NFT aggregates, as demonstrated by treating Tau mouse models with the autophagy inducer rapamycin or trehalose (Schaeffer *et al*, 2012; Caccamo *et al*, 2013; Ozcelik *et al*, 2013), TFEB targets both soluble and insoluble pTau species with no effect on endogenous Tau. Because soluble Tau is known to be neurotoxic (Santacruz *et al*, 2005), targeting only the aggregated Tau by enhancing autophagy is expected to offer limited therapeutic benefit. Furthermore, autophagy activation without enhanced lysosomal function will likely lead to the accumulation of autophagic cargoes detrimental to neuronal health. Indeed, extensive autophagic vacuoles can be observed in human AD brains and in transgenic mice expressing mutant P301L Tau (Lin *et al*, 2003; Nixon *et al*, 2005). TFEB in this regard dynamically regulates the ALP by coordinating autophagy induction with enhanced lysosomal clearance. As such, targeting TFEB-mediated cellular clearance may offer more superior therapeutic benefits than autophagy activators (reviewed in (Cuervo, 2011)).

In summary, we demonstrate that TFEB targets only the aberrant hyperphosphorylated and misfolded Tau while leaving the normal Tau intact; it is highly efficacious in ameliorating pTau/NFT pathology, neurodegeneration and behavioral deficits in rTg4510 mice while exhibiting no adverse effects on wild-type mice. These features make TFEB an attractive therapeutic target for AD and other diseases of tauopathy. However, the current work represents a proof-of-concept study. A TFEB-based therapy likely requires the identification of specific small molecule TFEB activators. Furthermore, it is important to note that, as a master regulator of lysosomal activity, expression levels and duration of TFEB need to be properly controlled. Indeed, dysregulation of members of the microphthalmia family of transcription factors including TFEB have been shown to cause renal carcinomas (Haq & Fisher, 2011). Therefore, rigorous studies are needed to evaluate the safety profiles of potential TFEB activators.

# Materials and Methods

## Antibodies

CP13, PHF1 and MC1 antibodies were kind gifts from P. Davies (Albert Einstein College of Medicine). Other antibodies were purchased from commercial sources as follows: AT8 (Pierce) and AT100 (Innogenetics), total Tau (DAKO), Tau1 (Millipore), PTEN (Cell Signaling), LAMP1 (Millipore), LC3 (Novus Biological and Sigma), Akt (Cell Signaling) and pAkt (Cell Signaling), P70S6K (Cell Signaling) and pP70S6K (Cell Signaling), ULK1 (Cell Signaling) and pULK1 (Cell Signaling), Y188 (Epitomics), 6E10 (Covance), NeuN (Chemicon), GFAP (DAKO), Iba1 (Waco), GAPDH (Sigma), FLAG (Biolegend), human TFEB (Cell Signaling), mouse and human TFEB (Abcam and Abmart), and γ-tubulin (Sigma).

## Animals

All procedures involving mice were approved by the Institutional Animal Care and Use Committee of the Baylor College of Medicine. The 5xFAD APP (Oakley *et al*, 2006) and rTg4510 Tau (Ramsden *et al*, 2005; Santacruz *et al*, 2005) transgenic mouse lines were obtained from the Jackson Laboratories and produced by crossing the APP transgenic mice with C57BL/6J mice, and the transactivator line CaMKIIα-tTA (on 129S6 background) with the Tau responder line (on FVB background), respectively. The littermate wild-type mice were used as controls. Both males and females were used. Mice within each genotype were randomly assigned for GFP or TFEB injections.

## Cell culture, transfection and luciferase assay

The double-stable Tet-On inducible Tau-expression cell line, T40PL, that expresses full-length human Tau with the P301L mutation (T40PL) in HEK293 cells under the regulation of a tightly controlled tetracycline-inducible promoter system (Invitrogen) and maintained in DMEM supplemented with 10% Tet-system-approved FBS (Clontech), 100 units/ml penicillin G, 100 μg/ml streptomycin, 400 μg/ml G418 (Sigma) + 0.20 μg/ml Puromycin (Clontech) at 37°C with 5% $CO_2$. Cells were transfected with 3xFLAG-tagged human TFEB plasmid, PTEN expression vector or PTEN shRNA using X-tremeGENE 9 DNA transfection reagent (Roche) Lipofectamine 3000 (GE lifesciences) according to the company's suggested protocol. Doxycycline (DOX) (Clontech) was added to the cell medium 24 h after TFEB transfection to a final concentration of 500 ng/ml. Cells were collected in ice-cold Tris-buffered saline (TBS, 50 mM Tris-HCl, pH 7.4, 150 mM NaCl) at various times after DOX induction as indicated.

N2a cells grown in 12-well plates were cotransfected with a TFEB-FLAG expression vector and PTEN promoter (−1334 to +1, −453 to +1 or −203 to +1)-luciferase reporters (Teresi *et al*, 2008), together with Renilla luciferase vector using X-tremeGENE 9 DNA Transfection Reagent (Roche). Twenty-four hours after transfection, cells were lysed with Passive Lysis Buffer (Promega). The Dual-Luciferase Reporter Assay System (Promega) was used to determine the firefly and Renilla luciferase activities according to the manufacturer's instructions. Measurements were performed with a BD luminometer, and firefly luciferase values were normalized to Renilla luciferase values. In all experiments, the internal control plasmid was used to compensate variable transfection efficiencies. All assays were repeated three times with each in triplicates.

## *In vivo* gene delivery

The AAV2/9-CMV-GFP (AAV-GFP), AAV2/9-CMV-TFEB (AAV-TFEB) or AAV2/9-CMV-TFEB-3xFLAG vectors were produced by the

TIGEM AAV Vector Core Facility as previously described (Settembre et al, 2011). For P0 injection, each mouse was injected into the lateral ventricles of both cerebral hemispheres with $4.2 \times 10^9$ total viral particles per side. The AAV-GFP and/or AAV-TFEB-injected wild-type, 5xFAD and rTg4510 mice were euthanized at 1 or 4 months after the injection. Adult rTg4510 Tau mice at 2 months of age were injected into the cortex and hippocampus (AP: + 1.5 mm, LAT: + 1.5 mm, DV: + 1.5 mm and AP: −2 mm, LAT: −1.5 mm, DV: + 1.75 mm) of both cerebral hemispheres according to the stereotaxic atlas of Franklin and Paxinos (Franklin & Paxinos, 2001) using the same total viral concentrations and analyzed also at 4 months.

### Microarray analysis

Mice were euthanized at 4 months of age, and hippocampi were dissected and frozen in liquid nitrogen immediately. Total RNA was isolated using RNeasy Lipid Tissue Mini Kit (Qiagen) and tested for quality assurance on the Aglient 2100 Bioanalyzer. For gene expression analysis, we used a Mouse Genome 430 2.0 Array from Affymetrix that provides coverage of the transcribed mouse genome in a single array (over 39,000 transcripts). Scanning was done using Affymetrix GeneChip Scanner 3000. Affymetrix labeling, hybridization, staining, washing, scanning and statistical analysis were done by the Microarray Core at Baylor College of Medicine (http://www.bcm.edu/mcfweb/). Accession number GSE53480.

Gene set enrichment analysis (GSEA) was performed as previously described (Subramanian et al, 2005). The cumulative distribution function was constructed by performing 1,000 random gene set membership assignments. A nominal P-value of < 0.01 and an FDR of < 10% were used to assess the significance of the enrichment score (ES). Gene Ontology (GO) analyses were performed with the web tool DAVID (da Huang et al, 2009) using default parameters. Redundant terms were manually removed from the resulting lists. The synaptic gene network was obtained by performing pathway co-expression analyses as previously described (Palmieri et al, 2011; Song et al, 2013) using genes annotated as 'Synapse' in the Gene Ontology database (The Gene Ontology Consortium, 2012). Briefly, 'Synapse' genes were used to analyze a vast set of transcriptional profiles available at the Gene Expression Omnibus (GEO) database (Barrett et al, 2013). Multiple cellular conditions and tissues are represented in this database. To ensure data homogeneity, the analysis was focused on experiments that used the Affymetrix platform Mouse Genome 430 2.0 Array. For each 'Synapse' gene pair, a pairwise co-expression score was calculated as their cumulative occurrence in the top 3% of correlated genes across all investigated experiments (Palmieri et al, 2011; Song et al, 2013). The expression correlation data were then analyzed with Cytoscape (Lopes et al, 2010) to draw a visual representation of expression relationships among genes. Downregulated and upregulated 'Synapse' genes in Tau and TFEB-injected Tau mice were highlighted in the Cytoscape-generated network by using different color codes.

### RNA extraction, reverse transcription and qRT-PCR

Total RNA was extracted from forebrain samples with TRIzol reagent (Invitrogen). Reverse transcription was performed using

SuperScript III reverse transcriptase reagents (Invitrogen). Quantitative PCR was done using Perfecta SYBR Green Fastmix (Quanta Biosciences) utilizing the ABI Prism 7000 detection system (Applied Biosystems). For expression studies, the qRT-PCR results were normalized against an internal control (GAPDH). Primers were designed with Primer Express Version 2.0 software (Applied Biosystems) using sequence data from NCBI. GAPDH primers were used as an internal control for each specific gene amplification. The relative levels of expression were quantified and analyzed by using ABI PRISM Sequence Detection System 7000 software. The real-time value for each sample was averaged and compared using the comparative CT method. The relative amount of target RNA was calculated relative to the expression of endogenous reference and relative to a calibrator which was the mean CT of control samples.

### Western blotting

For Westerns without fractionation, cells or forebrain tissues were lysed by RIPA buffer (TBS with 1% NP-40, 1% sodium deoxycholic acid, 0.1% sodium dodecylsulfate, and protease phosphatase inhibitor cocktails (Roche)). Cell lysates were sonicated for 6 pulses at 50% duty cycle, incubated at 4°C for 30 min and centrifuged at $20,000 \times g$ for 15 min. Supernatants were used for SDS-PAGE, transferred to PVDF membranes and detected using the ECL method (Pierce). Protein levels were quantified using ImageJ (National Institute of Health). In repeated experiments, we used infrared dye conjugated 2nd antibodies (IRDye 680RD anti-mouse IgG, IRDye 680RD anti-rat IgG and IRDye 800CW anti-rabbit IgG) in two-color combination and LI-COR Odyssey Imaging System for Western blot analysis and quantification. We changed from ECL as the detection method for Western blot to avoid overexposure issues; these near-infrared fluorescent 2nd antibodies result in a much large dynamic detection range than ECL.

For subcellular fractionation, cortex dissected from cerebral hemispheres of AAV-TFEB P0-injected mice were weighed, and homogenized by Dounce homogenizer in five volumes of TBS with 10% sucrose, protease inhibitor and phosphatase inhibitor cocktails. The mixture was centrifuged at $800 \times g$ for 5 min. Pellets were resuspended in the same volume of the above buffer and centrifuged again at the same speed. Supernatants from the two steps were pooled together and 1 ml of the solution from each mouse was subjected to ultracentrifugation at $100,000 \times g$ for 1 h. 700 µl of supernatants from ultracentrifugation were collected as the cytosol fractions. Pellets from the ultracentrifugation were resuspended in 700 µl of 1% sarkosyl in TBS with protease inhibitor and phosphatase inhibitor cocktails, sonicated, incubated in shaking for 30 min and ultracentrifuged again at $100,000 \times g$ for 1 h. Pellets from the second ultracentrifugation step were resuspended in 100 µl of 2x SDS-PAGE sample loading buffer, sonicated, and boiled for 5 min, and used as the sarkosyl-insoluble fractions. All the above procedures prior to boiling in SDS-sample loading buffer were carried out either on ice or in 4°C cold room. Either 5 µl of cytosol fractions, (1/140 of total) or 10 µl of insoluble fractions (1/10 of each sample) was loaded per gel for SDS-PAGE and the following Western blot analysis. All the cytosol fractions have protein concentration 25 ± 1.8 mg/ml by protein concentration assay using Biorad protein dye reagent (Biorad). No protein concentration assay was done for the insoluble fractions.

## Histology and Immunofluorescence

Cells grown on cover slips were fixed by 4% paraformaldehyde at room temperature for 10 min after removal of culture medium, washed in TBS for 5 min, followed by the same staining procedure as brain sections. Mice brains were collected after PBS perfusion; this procedure was followed by an over-night fixation of the brains with 4% paraformaldehyde in PBS. Then the brains were dehydrated in 30% sucrose in PBS. Immunofluorescence analyses were performed on 30-µm frozen sections. The slices were incubated for 1 h with TBS blocking solution before incubation overnight with the primary antibodies. After washing, the sections were incubated for 1 h with secondary antibodies conjugated either with Alexafluor 488 or Alexafluor 647 (Invitrogen). Images were taken on a confocal microscope (Leica SPE). The number of positive cells was quantified by using ImageJ software analysis. The number of autophagosomes per cell was quantified based on puncta number in the green channel. The number of total autophagosomes and autolysosomes per cell was quantified based on puncta number in the red channel. The number of autolysosomes per cell was calculated as total number minus number of autophagosomes.

## Quantitative colocalization analysis

DOX (1,000 µg/ml) was added to culture medium when T40PL cells were plated on cover slips in 24-well plates. 3xFLAG-TFEB and pEGFP (control) plasmids were transfected using Lipofecta-mine 3000 (GE lifesciences) 24 h after DOX induction. Cells on coverslips were fixed at 0, 8, 16, 24, 32 and 40 h time points for immunofluorescent staining. Cells transfected with 3xFLAG-TFEB were stained by antibodies anti-FLAG (rat mAb) and PHF1 (mouse mAb) antibodies, while cells transfected with pEGFP were stained by PHF1 only. To minimize the bleed-through effect in immunoflu-orescent image collection, Alexa-647-anti-mouse IgG (Invitrogen) was chosen to be paired with Alexa-488-anti-rat IgG as the secondary antibody for PHF1. Confocal images were collected on a Leica TCS SPE microscope; data analysis was carried out using Fiji-ImageJ (Schindelin *et al*, 2012). Specifically, for images collected using 10× objective lens, 8 confocal projection slices per view field, 4 view fields per time point per transfection were analyzed by plugin 'coloc2', in which the threshold of images were chosen automatically for objective analysis. Pearson Product Moment Correlation (also called Pearson correlation coefficient or Pearson R-value) above the threshold for each time point and transfection group were averaged and represented.

## CLEAR sequence analysis and chromatin immunoprecipitation (ChIP)

Human and mouse gene promoters were retrieved from the Ensemble database (http://www.ensembl.org) and searched with the CLEAR PWM using the PatSer tool with default parameters as described (Sardiello *et al*, 2009; Settembre *et al*, 2011).

The ChIP analysis was performed as previously described (Palmieri *et al*, 2011). Briefly, formaldehyde-fixed nuclei were isolated from HeLa transfectants carrying a TFEB-3xFLAG transgene or a control HeLa cell line without any tagged transgene. ChIP was performed using the ANTI-FLAG M2 Affinity Gel (Sigma) or IgG according

to the manufacturer's protocol. Each ChIP experiment required $10^7$ cells.

Oligonucleotide sequences are as follows:

HPRT-F: GCCACAGGTAGTGCAAGGTCTT; HPRT-R: TTCATGGC GGCCGTAAAC

APRT-F: GCCTTGACTCGCACTTTTGT; APRT-R: TAGGCGCCATC GATTTTAAG

MCOLN1-F: AGGGGCTCTGGGCTACC; MCOLN1-R: GCCCGCCG CTGTCACTG

PTEN-F: ATGTGGCGGGACTCTTTATG; PTEN-R: ACAGCGGCT-CAACTCTCAAAC

## Morris water maze (MWM) assay

MWM tests were performed in 4-month-old male mice. We used 54 animals in total for behavior test, at 10–14 mice/group. The genotypes and treatment groups were blinded to the experimenter. Briefly, mice were trained to locate a square platform (9 × 9 cm) hidden in Quadrant 3 (1–1.5 cm below water surface) of a circular water pool (1.38 m diameter, 21°C) within 60. In the learning sessions, each mouse was trained for 8 training blocks in four consecutive days. Each day, mice would swim for 4 successive trials as one training block before being returned to its holding cage. At the end of the 8th training block on day 4, mice were subjected to the probe test during which the platform was removed and each mouse would swim for 60 s to search for the platform. For all the trials, mice were allowed to climb to the platform and rest for 10 s before they were submitted to the next trial or trans-ferred back to holding cages. The time for mice to locate the hidden platform during learning, as well as the number of platform crossings during the probe tests, was recorded by Ethovision tracking system (Noldus Information Technologies).

## Electrophysiology

Field recordings of Schaffer collateral LTP was performed as described (Peethumnongsin *et al*, 2010). Briefly, brains were isolated from 4-month-old mice and cut into 400-µM slices (Leica). Hippocampal slices were incubated for 1 h at room temperature and then transferred to a heated recording chamber filled with recording ACSF (125 mM NaCl, 2.5 mM KCl, 1.25 mM $NaH_2PO_4$, 25 mM $NaHCO_3$, 1 mM $MgCl_2$, 2 mM $CaCl_2$, and 10 mM glucose, saturated with 95% $O_2$ and 5% $CO_2$) maintained at 32°C. Stimulation of Schaffer collaterals from the CA3 region was performed with bipolar electrodes, while borosilicate glass capillary pipettes filled with recording ACSF (resistances of 2–3.5 M$\Omega$) were used to record field excitatory postsynaptic potentials (fEPSPs) in the CA1 region. Signals were amplified using a MultiClamp 700 B amplifier (Axon), digitized using a Digidata 1322A (Axon) with a 2 kHz low pass filter and a 3 Hz high pass filter and then captured and stored using Clampex 9 software (Axon) for offline data analysis. The genotypes and treatment groups were blinded to the experimenter.

## Stereology

The number of CA1 neurons was assessed on NeuN-DAB stained sections from wild-type and rTg4510 Tau mice with either AAV-GFP or AAV-TFEB P0 injections (3 in each genotype and treatment group) at 4 months of age. Unbiased stereology was performed using optical

**The paper explained**

**Problem**

Alzheimer's disease has two pathological hallmarks: β-amyloid plaques and neurofibrillary tangles (NFTs), the latter are composed of misfolded Tau protein. In the past, the field has devoted intensive effort on β-amyloid based therapies and these have yielded disappointing results so far. One of the concerns may is that these agents may not target the NFT pathology and strong evidence has implicated a neurotoxic function of NFTs in Alzheimer's disease and other neurodegenerative diseases. Therefore, there is increasing interest in developing Tau-based therapy for treating Alzheimer's disease.

**Results**

In the current study, we investigated the role of Transcription Factor EB (TFEB), a molecule shown to play central roles in the autophagy and lysosomal pathway, in the rTg4510 Tau transgenic mouse models. We found that adeno-associated virus-mediated TFEB expression has no untoward side effects on wild-type mice. In contrast, TFEB is highly efficacious in reducing neurofibrillary tangle pathology and rescuing behavioral and synaptic deficits and neurodegeneration in the Tau mouse model. TFEB specifically targets the misfolded Tau species while leaving normal Tau intact. We provide evidence that this effect requires lysosomal activity and we identify phosphatase and tensin homolog (PTEN) as a direct target of TFEB that may be required for TFEB-dependent aberrant Tau clearance.

**Impact**

The specificity and efficacy of TFEB in mediating the clearance of toxic Tau species and rescuing neurodegeneration is remarkable. Our findings provide proof-of-concept that small molecule TFEB activators may serve as effective therapy for treating diseases of tauopathy including Alzheimer's disease.

## Acknowledgments

We are grateful to Dr. P. Davies (Albert Einstein School of Medicine) for the gift of CP13, MC1 and PHF1 antibodies, Dr. C. Eng (Cleveland Clinic Foundation) for providing the pGL3-PTEN reporter constructs and Dr. X. Jiang (Memorial Sloan Kettering Cancer Center) for the PTEN expression and shRNA constructs. We are indebted to C. Spencer and the Baylor College of Medicine IDDRC Administrative, Genomic and RNA Profiling and Mouse Neurobehavioral cores (HD024064) for the generous resources and support. We thank A. Cole and N. Aithmitti for expert technical assistance and members of the Zheng laboratory for constructive discussions. This work was supported by grants from NIH (AG020670, AG032051 and NS076117 to HZ) and the Belfer Neurodegeneration Consortium.

## Author contributions

VAP, HL, HM-S, BW, LY, YX, DW, MP, AdR performed experiments and analyzed the data, VAP, HL and HM-S and HZ designed the study, MS performed bioinformatics analysis, VML and AB provided the Tau inducible cell line and TFEB vectors/viruses, respectively, as well as overall guidance, HZ wrote the manuscript together with VAP. All authors made comments on the manuscript.

## Conflict of interest

The authors declare that they have no conflict of interest.

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

dissector method (West & Gundersen, 1990) on Zeiss microscope with motorized stage. Data acquisition and analysis was performed using software StereoInvestigator by MBF Bioscience. The CA1 region of counting was outlined under 5× objective lens on every 10th section of 35-μm-thick cryostat sections from 0.24 to 1.36 mm lateral from the midline on one hemisphere of the mouse brain, according to the stereotaxic coordinates adopted from a mouse brain atlas (Franklin & Paxinos, 2001). NeuN-positive neurons were counted in counting frame area of 625 μm$^2$ with sampling grid area of 26,338 μm$^2$ under 40× objective lens. Volume of the region was determined by Cavalieri's method (Cruz-Orive, 1999).

## Statistics

All data are presented as average ± s.e.m. Power analysis was performed using confident interval of α = 0.05. $N$ = 3–14 mice/group were used for each experiment as specified. Outliers were identified using Grubbs' method with α = 0.05. Pairwise comparisons were analyzed using a two-tailed Student's *t*-test, while a two-way ANOVA with Bonferroni post hoc was used for multiple comparisons. Correlation between TFEB protein levels and PHF-1 was determined using Pearson Product Moment Correlation test. *P*-values less than or equal to 0.05 were considered statistically significant. *$P < 0.05$; **$P < 0.01$; *** $P < 0.001$.

**Supplementary information** for this article is available online: http://embomolmed.embopress.org

amyloid load, predict cognitive status in Alzheimer's disease. *Neurology* 60: 1495–1500

Haq R, Fisher DE (2011) Biology and clinical relevance of the micropthalmia family of transcription factors in human cancer. *J Clin Oncol* 29: 3474–3482

Hardy J (2006) A hundred years of Alzheimer's disease research. *Neuron* 52: 3–13

Harris H, Rubinsztein DC (2012) Control of autophagy as a therapy for neurodegenerative disease. *Nat Rev Neurol* 8: 108–117

da Huang W, Sherman BT, Lempicki RA (2009) Systematic and integrative analysis of large gene lists using DAVID bioinformatics resources. *Nat Protoc* 4: 44–57

Inoue K, Rispoli J, Kaphzan H, Klann E, Chen EI, Kim J, Komatsu M, Abeliovich A (2012) Macroautophagy deficiency mediates age-dependent neurodegeneration through a phospho-tau pathway. *Mol Neurodegener* 7: 48

Kerr F, Rickle A, Nayeem N, Brandner S, Cowburn RF, Lovestone S (2006) PTEN, a negative regulator of PI3 kinase signalling, alters tau phosphorylation in cells by mechanisms independent of GSK-3. *FEBS Lett* 580: 3121–3128

Kimura S, Noda T, Yoshimori T (2007) Dissection of the autophagosome maturation process by a novel reporter protein, tandem fluorescent-tagged LC3. *Autophagy* 3: 452–460

Kopeikina KJ, Hyman BT, Spires-Jones TL (2012) Soluble forms of tau are toxic in Alzheimer's disease. *Transl Neurosci* 3: 223–233

Kruger U, Wang Y, Kumar S, Mandelkow EM (2012) Autophagic degradation of tau in primary neurons and its enhancement by trehalose. *Neurobiol Aging* 33: 2291–2305

Kwon CH, Luikart BW, Powell CM, Zhou J, Matheny SA, Zhang W, Li Y, Baker SJ, Parada LF (2006) Pten regulates neuronal arborization and social interaction in mice. *Neuron* 50: 377–388

Kwon CH, Zhu X, Zhang J, Baker SJ (2003) mTor is required for hypertrophy of Pten-deficient neuronal soma in vivo. *Proc Natl Acad Sci USA* 100: 12923–12928

Kwon CH, Zhu X, Zhang J, Knoop LL, Tharp R, Smeyne RJ, Eberhart CG, Burger PC, Baker SJ (2001) Pten regulates neuronal soma size: a mouse model of Lhermitte-Duclos disease. *Nat Genet* 29: 404–411

Lin WL, Lewis J, Yen SH, Hutton M, Dickson DW (2003) Ultrastructural neuronal pathology in transgenic mice expressing mutant (P301L) human tau. *J Neurocytol* 32: 1091–1105

Lopes CT, Franz M, Kazi F, Donaldson SL, Morris Q, Bader GD (2010) Cytoscape Web: an interactive web-based network browser. *Bioinformatics* 26: 2347–2348

Mandelkow EM, Mandelkow E (2012) Biochemistry and cell biology of tau protein in neurofibrillary degeneration. *Cold Spring Harb Perspect Med* 2: a006247

Medina DL, Fraldi A, Bouche V, Annunziata F, Mansueto G, Spampanato C, Puri C, Pignata A, Martina JA, Sardiello M *et al* (2011) Transcriptional activation of lysosomal exocytosis promotes cellular clearance. *Dev Cell* 21: 421–430

Mueller-Steiner S, Zhou Y, Arai H, Roberson ED, Sun B, Chen J, Wang X, Yu G, Esposito L, Mucke L *et al* (2006) Antiamyloidogenic and neuroprotective functions of cathepsin B: implications for Alzheimer's disease. *Neuron* 51: 703–714

Mullard A (2012) Sting of Alzheimer's failures offset by upcoming prevention trials. *Nat Rev Drug Discov* 11: 657–660

Nixon RA (2007) Autophagy, amyloidogenesis and Alzheimer disease. *J Cell Sci* 120: 4081–4091

Nixon RA, Wegiel J, Kumar A, Yu WH, Peterhoff C, Cataldo A, Cuervo AM (2005) Extensive involvement of autophagy in Alzheimer disease: an immuno-electron microscopy study. *J Neuropathol Exp Neurol* 64: 113–122

Nixon RA, Yang DS (2011) Autophagy failure in Alzheimer's disease–locating the primary defect. *Neurobiol Dis* 43: 38–45

Oakley H, Cole SL, Logan S, Maus E, Shao P, Craft J, Guillozet-Bongaarts A, Ohno M, Disterhoft J, Van Eldik L *et al* (2006) Intraneuronal beta-amyloid aggregates, neurodegeneration, and neuron loss in transgenic mice with five familial Alzheimer's disease mutations: potential factors in amyloid plaque formation. *J Neurosci* 26: 10129–10140

Ozcelik S, Fraser G, Castets P, Schaeffer V, Skachokova Z, Breu K, Clavaguera F, Sinnreich M, Kappos L, Goedert M *et al* (2013) Rapamycin Attenuates the Progression of Tau Pathology in P301S Tau Transgenic Mice. *PLoS ONE* 8: e62459

Palmieri M, Impey S, Kang H, di Ronza A, Pelz C, Sardiello M, Ballabio A (2011) Characterization of the CLEAR network reveals an integrated control of cellular clearance pathways. *Hum Mol Genet* 20: 3852–3866

Peethumnongsin E, Yang L, Kallhoff-Munoz V, Hu L, Takashima A, Pautler RG, Zheng H (2010) Convergence of presenilin- and tau-mediated pathways on axonal trafficking and neuronal function. *J Neurosci* 30: 13409–13418

Pickford F, Masliah E, Britschgi M, Lucin K, Narasimhan R, Jaeger PA, Small S, Spencer B, Rockenstein E, Levine B *et al* (2008) The autophagy-related protein beclin 1 shows reduced expression in early Alzheimer disease and regulates amyloid beta accumulation in mice. *J Clin Invest* 118: 2190–2199

Ramsden M, Kotilinek L, Forster C, Paulson J, McGowan E, SantaCruz K, Guimaraes A, Yue M, Lewis J, Carlson G *et al* (2005) Age-dependent neurofibrillary tangle formation, neuron loss, and memory impairment in a mouse model of human tauopathy (P301L). *J Neurosci* 25: 10637–10647

Roczniak-Ferguson A, Petit CS, Froehlich F, Qian S, Ky J, Angarola B, Walther TC, Ferguson SM (2012) The transcription factor TFEB links mTORC1 signaling to transcriptional control of lysosome homeostasis. *Sci Signal* 5: ra42

Rohn TT, Wirawan E, Brown RJ, Harris JR, Masliah E, Vandenabeele P (2011) Depletion of Beclin-1 due to proteolytic cleavage by caspases in the Alzheimer's disease brain. *Neurobiol Dis* 43: 68–78

Santacruz K, Lewis J, Spires T, Paulson J, Kotilinek L, Ingelsson M, Guimaraes A, DeTure M, Ramsden M, McGowan E *et al* (2005) Tau suppression in a neurodegenerative mouse model improves memory function. *Science* 309: 476–481

Sardiello M, Palmieri M, di Ronza A, Medina DL, Valenza M, Gennarino VA, Di Malta C, Donaudy F, Embrione V, Polishchuk RS *et al* (2009) A gene network regulating lysosomal biogenesis and function. *Science* 325: 473–477

Schaeffer V, Lavenir I, Ozcelik S, Tolnay M, Winkler DT, Goedert M (2012) Stimulation of autophagy reduces neurodegeneration in a mouse model of human tauopathy. *Brain* 135: 2169–2177

Schindelin J, Arganda-Carreras I, Frise E, Kaynig V, Longair M, Pietzsch T, Preibisch S, Rueden C, Saalfeld S, Schmid B *et al* (2012) Fiji: an open-source platform for biological-image analysis. *Nat Methods* 9: 676–682

Settembre C, Di Malta C, Polito VA, Garcia Arencibia M, Vetrini F, Erdin S, Erdin SU, Huynh T, Medina D, Colella P *et al* (2011) TFEB links autophagy to lysosomal biogenesis. *Science* 332: 1429–1433

Settembre C, Zoncu R, Medina DL, Vetrini F, Erdin S, Huynh T, Ferron M, Karsenty G, Vellard MC, Facchinetti V *et al* (2012) A lysosome-to-nucleus signalling mechanism senses and regulates the lysosome via mTOR and TFEB. *EMBO J* 31: 1095–1108

Song MS, Salmena L, Pandolfi PP (2012) The functions and regulation of the PTEN tumour suppressor. *Nat Rev Mol Cell Biol* 13: 283−296

Song W, Wang F, Savini M, Ake A, di Ronza A, Sardiello M, Segatori L (2013) TFEB regulates lysosomal proteostasis. *Hum Mol Genet* 22: 1994−2009

Sperow M, Berry RB, Bayazitov IT, Zhu G, Baker SJ, Zakharenko SS (2011) Phosphatase and tensin homologue (PTEN) regulates synaptic plasticity independently of its effect on neuronal morphology and migration. *J Physiol* 590: 777−792

Subramanian A, Tamayo P, Mootha VK, Mukherjee S, Ebert BL, Gillette MA, Paulovich A, Pomeroy SL, Golub TR, Lander ES *et al* (2005) Gene set enrichment analysis: a knowledge-based approach for interpreting genome-wide expression profiles. *Proc Natl Acad Sci USA* 102: 15545−15550

Takeuchi K, Gertner MJ, Zhou J, Parada LF, Bennett MV, Zukin RS (2013) Dysregulation of synaptic plasticity precedes appearance of morphological defects in a Pten conditional knockout mouse model of autism. *Proc Natl Acad Sci USA* 110: 4738−4743

Teresi RE, Planchon SM, Waite KA, Eng C (2008) Regulation of the PTEN promoter by statins and SREBP. *Hum Mol Genet* 17: 919−928

The Gene Ontology Consortium (2012) Gene Ontology annotations and resources. *Nucleic Acids Res* 41: D530−D535

Wang X, Trotman LC, Koppie T, Alimonti A, Chen Z, Gao Z, Wang J, Erdjument-Bromage H, Tempst P, Cordon-Cardo C *et al* (2007) NEDD4-1 is a proto-oncogenic ubiquitin ligase for PTEN. *Cell* 128: 129−139

Wang Y, Martinez-Vicente M, Kruger U, Kaushik S, Wong E, Mandelkow EM, Cuervo AM, Mandelkow E (2009) Tau fragmentation, aggregation and clearance: the dual role of lysosomal processing. *Hum Mol Genet* 18: 4153−4170

Wang Y, Martinez-Vicente M, Kruger U, Kaushik S, Wong E, Mandelkow EM, Cuervo AM, Mandelkow E (2010) Synergy and antagonism of macroautophagy and chaperone-mediated autophagy in a cell model of pathological tau aggregation. *Autophagy* 6: 182−183

West MJ, Gundersen HJ (1990) Unbiased stereological estimation of the number of neurons in the human hippocampus. *J Comp Neurol* 296: 1−22

Yang DS, Stavrides P, Mohan PS, Kaushik S, Kumar A, Ohno M, Schmidt SD, Wesson D, Bandyopadhyay U, Jiang Y *et al* (2011) Reversal of autophagy dysfunction in the TgCRND8 mouse model of Alzheimer's disease ameliorates amyloid pathologies and memory deficits. *Brain* 134: 258−277

Zhang X, Li F, Bulloj A, Zhang YW, Tong G, Zhang Z, Liao FF, Xu H (2006a) Tumor-suppressor PTEN affects tau phosphorylation, aggregation, and binding to microtubules. *FASEB J* 20: 1272−1274

Zhang X, Zhang YW, Liu S, Bulloj A, Tong GG, Zhang Z, Liao FF, Xu H (2006b) Tumor suppressor PTEN affects tau phosphorylation: deficiency in the phosphatase activity of PTEN increases aggregation of an FTDP-17 mutant Tau. *Mol Neurodegener* 1: 7

Zhou J, Blundell J, Ogawa S, Kwon CH, Zhang W, Sinton C, Powell CM, Parada LF (2009) Pharmacological inhibition of mTORC1 suppresses anatomical, cellular, and behavioral abnormalities in neural-specific Pten knock-out mice. *J Neurosci* 29: 1773−1783

