## [Review Process File · EMBO Molecular Medicine]

Selective Clearance of Aberrant Tau Proteins and Rescue of Neurotoxicity by Transcription Factor EB

Vinicia Assunta Polito, Hongmei Li, Heidi Martini-Stoica, Baiping Wang, Li Yang, Yin Xu, Daniel B. Swartzlander, Michela Palmieri, Alberto di Ronza, Virginia M.-Y. Lee, Marco Sardiello, Andrea Ballabio and Hui Zheng

Corresponding author: Hui Zheng, Baylor College of Medicine

Review timeline:

Submission date:	14 November 2013
Editorial Decision:	09 December 2013
Revision received:	15 April 2014
Editorial Decision:	13 May 2014
Revision received:	09 June 2014
Editorial Decision:	23 June 2014
Revision received:	26 June 2014
Accepted:	30 June 2014

Transaction Report:

Editor: Céline Carret

1st Editorial Decision

09 December 2013

Thank you for the submission of your manuscript to EMBO Molecular Medicine. We have now heard back from the three referees whom we asked to evaluate your manuscript. Although the referees find the study to be potentially important, they also raise serious concerns that should be addressed in a major revision of the study.

As you will see from the below pasted comments, all three referees are enthusiastic about the significance and interest of the findings. However, all three are also concerned about the limited mechanism provided. While referee 2 agrees that a detailed molecular mechanism would be beyond the topic of this study, additional experiments are required to improve the conclusiveness of the data and provide a better understanding of the underlying process. In short, better western blots should be provided and results quantified, important controls are missing and some clarifications/explanations are necessary as highlighted by the referees.

Given these evaluations, I would like to give you the opportunity to revise your manuscript, with the understanding that the referees' concerns must be fully addressed and that acceptance of the manuscript would entail a second round of review. Please note that it is our journal's policy to allow only a single round of revision, and that acceptance or rejection of the manuscript will therefore depend on the completeness of your response and the satisfaction of the referees with it.

Please see below for additional editorial requirements.

I look forward to seeing a revised form of your manuscript as soon as possible.

***** Reviewer's comments *****

Referee #1 (Comments on Novelty/Model System):

There are other more severe tauopathy mice models available, which could perhaps further attest to the strength of their findings in a follow-up manuscript.

Referee #1 (Remarks):

This manuscript by Polito & Li et al describes selective clearance of misfolded tau protein by TFEB overexpression. The authors also report a novel target for TFEB transcription, the lipid phosphatase PTEN, and present evidence for a mechanistic link between TFEB overexpression, PTEN induction and macroautophagy-independent misfolded Tau clearance. The novelty of this manuscript lies in that this is the first evidence of TFEB-upregulation as a therapeutic target in AD/tauopathies, the identification of PTEN as a new TFEB target, and the finding that long-term TFEB expression is not deleterious in mouse brain.

While the conclusions are significant for the field of TFEB and neurodegeneration research, more evidence supporting the authors' hypothesis is required before their story is completely sound. Mainly, the authors' elimination of macroautophagy as responsible for TFEB-reduction of phospho-tau is incomplete and needs further data to support this conclusion. There is a chronic lack of western blot densitometry that needs to be remedied in order to properly quantify observed effects. One issue that must be addressed is that the authors' final statement that 'TFEB was effective when introduced both before and after the onset of neuropathology' is incorrect. While they do indeed report both pathology and phenotype rescue in their P0 injected Tau mice, only Tau staining is shown to be reduced in 2M injected mice. No behavior rescue is reported for after-onset treatment. Thus, care should be taken in making such broad statements regarding therapy development.

Other specific concerns that need to be addressed regarding each of the figures are as follows:

Figure 1:

1A) The authors do not report expression pattern or levels of TFEB transduction after injection in P0 mice. Whereas they show very nice coverage with their AAV-GFP control vector (Suppl Figure 1), no TFEB staining is shown. This confounds their results, as it is impossible to determine expression levels in relevant brain regions or cell-to-cell analysis for decreases in tau.

1C) No western blot of TFEB expression is shown, which is necessary to compare TFEB expression levels between Wt and Tau mice. This is particularly important as the qRT-PCR for TFEB expression levels shown in Suppl Figure 1 does seem to indicate differential expression of TFEB in both lines of mice. Since such data is shown for 2 month-old injected mice, it is unclear why this was not analyzed for P0 mice as well. Furthermore, authors state that there are no differences in total tau protein levels in Tau mice after TFEB injection. However, western blot shown is overexposed and non-quantifiable. Furthermore, variability of triplicates shown on the western blot in 2C does not reflect the tight error bars shown in the quantification graph in 2D, particularly for AT8 and MC1 probing.

1F) Quantification of TFEB reduction of insoluble Tau levels is missing.

Figure 2:

2A) No GFP channel is shown for control injected mice, which makes it difficult to compare reduced Tau staining in TFEB-only expressing cells.

2B) Again, total Tau and tubulin western blots shown are overexposed.

Figure 3: Authors claim that their effect is cell-autonomous based on inducible cell-line results shown in Figure 2. However, in this *in vivo* pathology rescue experiment, they do not show TFEB staining channel to confirm that surviving NeuN+ positive cells are indeed also TFEB+. This is particularly important as we also see reduced astroglial/inflammatory response in Figure 3C-F, suggesting a non-autonomous astrocyte-mediated response could also underlie improved neuronal survival. Quantification of cell number, particularly in the affected CA1 area of the hippocampus, with and without TFEB would also be desirable. Also, is neuronal survival improved in 2M old injections? Brain weight is not a direct measure of neuronal survival, so 3F needs to be supported by similar analysis as done in P0 injected mice in 3A.

Figure 4: Synaptic alterations in these Tau mice have already been reported (Kopeikina et al 2013). Authors do not show comparison between Wt and Wt + TFEB in their microarray analysis in 4E.

Figure 5: Based on their microarray analysis, the authors propose that, *in vivo*, TFEB upregulation leads to a coordinated early up-regulation of autophagy followed by a chronic up-regulation of lysosomal-mediated clearance. While this is particularly interesting for the field of TFEB and autophagy research, and deserves to be expanded more, the functional characterization of this possible pathway is not explored properly. Only readout of lysosomal clearance shown is increased LAMP1 protein levels on a western blot (and the effect is quite modest), and this does not reflect functional upregulation of lysosomes. Additional experiments are required for this conclusion. Furthermore, the conclusion that since there is 'no appreciable transcriptional regulation of the autophagy pathway' after TFEB injection means there is no functional autophagy induction is an over-interpretation of their results. While indeed they see no significant expression changes of autophagy genes after TFEB induction, the autophagy pathway functions mostly by activating already synthesized-proteins, via post-translational modifications or sub-cellular relocalization. Thus, in order to conclude that autophagy is indeed not responsible for decreased levels of insoluble tau, the authors need to do more exhaustive analysis of their *in vivo* data, such as p62 or LC3 puncta staining (and their colocalization, or lack thereof, with tau aggregates) in hippocampal neurons +/- TFEB injections. Processing and conjugation of other Atg proteins (such as Atg5-Atg12 or Beclin-1) is also an option. Finally, to determine if this effect is indeed independent of macroautophagy, blocking macroautophagy by Atg5/7 knockdown or spautin treatment should also be considered.

Figure 7:

7A) It is unclear what cells were used for this assay. Please add quantification of western blot data. While the effect of TFEB overexpression on AKT inhibition seems to be quite significant, the 'associated reduced pP70SK6 and pULK1' are very modest and require densitometry analysis. Overexposure of bands is also an issue in this figure panel.

7B) PTEN overexpression is running higher than endogenous PTEN. Is the transfected PTEN tagged with something or is this just a nonspecific band? Furthermore, authors use this figure to establish that TFEB-reduction in levels of P-Tau is independent of PTEN phosphatase activity on Tau protein. However, glaring overexposure of the unphosphorylated (Tau1) and total Tau blots makes it impossible to make this determination. Authors should re-expose and quantify these blots. Additionally, use of a PTEN-insensitive Tau or a phosphatase-dead PTEN are needed in this experiment (or in 7C) to verify that reductions in phospho-Tau levels after TFEB expression are not due to PTEN dephosphorylating tau.

Referee #2 (Comments on Novelty/Model System):

While it is clear that TFEB over-expression has an effect in the model of tauopathy that was studied, the lack of information concerning levels and localization of the over-expressed protein is a significant concern both for interpretation of the data and for ensuring that other labs could understand the model well enough to replicate such findings.

With respect to the effect of TFEB over-expression on amyloid plaque abundance in the 5xFAD mouse model, the conclusion is that there is no effect on plaque density. However, the representative images that were provided raised concerns about the staining and identification of the plaques. Addressing these and other concerns that I have outlined below would greatly improve the technical quality of the manuscript.

The lack of a clearly defined mechanism for how TFEB over-expression reduces the abundance of neurofibrillary tangles reduced evaluation of the medical impact. I recognize though that this would be a difficult issue to address.

Referee #2 (Remarks):

This manuscript investigates the effects of over-expression of transcription factor EB (TFEB) on two distinct mouse models of Alzheimer's disease pathology that exhibit amyloid plaques and neurofibrillary tangles respectively. The major findings reported by the authors are that TFEB over-expression reduces neurofibrillary tangle pathology in a manner that appears to depend on the TFEB-dependent up-regulation of PTEN expression. While the effects of TFEB over-expression on reducing tauopathy are impressive, the mechanisms whereby such effects are achieved, including the relevant targets downstream of PTEN, remain unclear. Elucidating such mechanisms in detail is likely beyond the scope of this manuscript. Therefore, the concerns presented below focus specifically on interpretation of the effects of TFEB over-expression on disease pathology.

Specific Concerns

1. The authors characterize their TFEB over-expression strategy as "mild Adeno-Associated Virus (AAV)-mediated TFEB expression". However, no data is provided that compares the endogenous to the over-expressed TFEB protein levels. A 2-3 fold change was seen at mRNA level (Fig. S1). Was this for whole brain? Based on GFP expression pattern in Fig S1, the effect would likely be much greater in the cerebral cortex. As the findings of the paper depend so heavily on this TFEB over-expression strategy, a better characterization of the TFEB protein levels in relevant brain regions of control versus TFEB over-expressing mice is critical. Such a characterization should include measurement of TFEB protein levels as well as an assessment of the fraction of neurons that over-express TFEB in the relevant brain regions.
2. Fig. S2. It is concluded based on the data presented in Fig. S2 that TFEB over-expression does not affect amyloid plaque abundance in the 5xFAD mouse model of Alzheimer's disease. However, the punctate staining in the images of APP antibody stained brain sections does not match the expected appearance of amyloid plaque staining in this mouse line as there are no plaques evident in the hippocampi or in superficial layers of the cerebral cortices. Thus, it is not certain that the staining in these images actually represents plaques. Analysis of staining with an additional marker of plaques such as thioflavin S is required to ensure that amyloid plaques have been properly identified in these experiments.
3. No evidence is provided to demonstrate that robust TFEB over-expression (at the protein level) was successfully achieved in the 5xFAD mice. This limits the confidence concerning the interpretation of the negative result relating to plaque abundance.
4. In figure 1C and D, the representative blots show considerable variability yet the accompanying graphs indicate practically none. This discrepancy should be explained.

5. Figure 2 appears to compare TFEB/GFP AAV injected brains to untreated controls. The ideal control would have been injection of a comparable titer of GFP AAV. A justification for this less than ideal control should be provided.
6. For the experiments in Figure 2, is this a TFEB-GFP fusion protein or the same untagged TFEB construct that was used elsewhere?
7. What was the level of TFEB overexpression (protein levels) in the T40PL cell line that was used to achieve the effects shown in Figure 2D?
8. Over-expression of TFEB (and closely related transcription factors) causes renal carcinoma in humans (PMID: 21670463). This link between TFEB and cancer raises concerns about the use of therapeutic strategies that would intentionally raise the expression of this protein. As the findings of the current manuscript have the potential to raise the hopes and expectations of patients and/or the families of patients who suffer from Alzheimer's disease, it is important that such caveats are also acknowledged.

Referee #3 (Remarks):

This is an interesting and potentially significant set of findings on the ability of AAV-mediated TFEB expression to prevent or rescue diverse aspects of disease in a mouse model of tauopathy. The data demonstrate that PTEN is a direct target of TFEB, which is required for TFEB dependent ptau clearance. The authors persuasively demonstrate that TFEB expression reduces levels of phosphorylated and misfolded tau species. They conclude that the mechanism involves a selective targeting of these altered tau species in both soluble and insoluble form to the autophagy pathway where they are degraded within lysosomes. The selective targeting is somewhat unanticipated given previous studies indicating that tau aggregates are a selective target for autophagy and other studies suggesting that all forms of tau, especially when overexpressed, may be substrates for autophagy. The major concern in the study is that, while it is possible that pathogenic forms of tau are selectively targeted by TFEB mediated autophagy, the evidence that the observed changes on tau are in fact autophagy and lysosomal dependent is not firmly supported by the data presented. In addition, there is no clear mechanism to account for the selectivity either demonstrated experimentally or discussed that would gain support from the known effects of TFEB demonstrated in this system. These issues could be addressed experimentally as discussed below.

1. The evidence that TFEB is altering tau through an autophagy/lysosomal-dependent mechanism is mainly based on the previous work by these authors, and evidence presented in the report establishing a lysosomal-dependent mechanism for tau changes is circumstantial. The strongest case, which is still indirect, is made by the cell studies in which total tau levels are decreased in both soluble and insoluble fractions. Further studies are needed to demonstrate lysosomal dependence, such as analyses involving the clamping of lysosomal proteolysis with an inhibitor and/or demonstrating the presence of the tau substrates in lysosomes and ideally also directly demonstrating that lysosomal activity is enhanced as predicted. It is not clear from the data presentation whether the mechanism proposed is solely related to an enhancement of lysosomal activity rather than, or in addition to, a change in autophagy induction. The discrimination of these two possible mechanisms is important since it is not clear how the authors would explain the selective elimination of phosphotau species by only an enhancement of lysosomal activity, except possibly through CMA.

2. The *in vivo* studies are subject to the same concerns, which are heightened by the fact that total tau levels are not changing after TFEB expression. This observation is interpreted as reflecting selective degradation of phosphotau and misfolded tau species although observed tau changes could alternatively be explained by alterations of tau phosphorylation. The authors attempted to exclude the possibility of a change in phosphorylation by performing western blot analyses of relevant

protein kinases and phosphatases but this analysis is minimally informative because the phosphorylation states of the protein kinases, which are the only appropriate way to assess kinase activation changes using immunoblot analysis, were not monitored. In the case of cdk5, p25 levels need to be measured to assess activation. The panel of possible tau phosphatases analyzed is also incomplete. Based on the collective data in this report, therefore, the possibility that these changes relate to effects of TFEB expression on tau phosphorylation cannot be excluded. Shifts in phosphorylation could well account for differential effects on distributions of phospho and total tau species in the soluble and insoluble pools. In the absence of direct evidence for a lysosomal-dependent pathway and a mechanism for selective targeting for phosphotau species to lysosomes, the possible effects on phosphatases or kinases has to be excluded rigorously.

3. There are several concerns about the immunoblot analyses of tau fractions which further hinder the evaluation and interpretation of these results. There is the general concern that none of these data are quantified and some involve very small numbers of animals. Information is not evident about how much of each tau fraction was loaded onto gels or whether immunoblots of different fractions can be directly compared (i.e., run on the same gel or exposed for equal times). This results in what may be a misleading impression that insoluble and soluble tau are equally abundant. Without information about the relative abundance of these tau forms, it is difficult to interpret, for example, how a change in insoluble tau forms may or may not impact the soluble pool and whether changes in insoluble forms reflect proteolytic loss rather than disaggregation and redistribution to a soluble pool. The choice of gamma tubulin as a loading control, rather than a more conventional tubulin species or another protein, is unclear and the result presented gives the impression that insoluble and soluble gamma tubulin are equally abundant, which would be surprising.

4. If it could be established firmly that ptau is selectively targeted to an autophagy system, this finding and the functional rescue of the animals by TFEB, are sufficiently exciting with or without a clarification of the targeting mechanism, but it would be helpful to have a more rigorous discussion of the possible mechanisms for selective targeting of ptau, including how a CMA mechanism might operate and also be consistent with the current data.

Additional comments/questions

1. For at least some of the tau immunoblot data, a presentation of the entire length of the gel should be included to allow evaluation of additional tau species, including breakdown products, which would be expected to be present and potentially informative.
2. It is not clear which supernatant fraction is loaded in Figure 1E.
3. The tau Dako antibody, used in this study as a measure of total tau, is stated in the Dako website to be recognizing non-phosphorylating tau.
4. Ages of APP mice analyzed should be provided.

1st Revision - authors' response

15 April 2014

We would like to thank the editor for the prompt handling of our manuscript "Selective Clearance of Aberrant Tau Proteins and Rescue of Neurotoxicity by Transcription Factor EB," and the reviewers for their comprehensive and insightful reviews, which we have taken to heart. We are pleased that the editor and the referees are enthusiastic about the novelty and significance of our findings. In this extensively revised manuscript, we have included additional data and provided adequate controls and quantifications as requested by the reviewers. Below are the reviewers' critiques (in *italics*) followed by our point-by-point responses.

Referee #1:

This manuscript by Polito & Li et al describes selective clearance of misfolded tau protein by TFEB overexpression. The authors also report a novel target for TFEB transcription, the lipid

phosphatase PTEN, and present evidence for a mechanistic link between TFEB overexpression, PTEN induction and macroautophagy-independent misfolded Tau clearance. The novelty of this manuscript lies in that this is the first evidence of TFEB-upregulation as a therapeutic target in AD/tauopathies, the identification of PTEN as a new TFEB target, and the finding that long-term TFEB expression is not deleterious in mouse brain.

While the conclusions are significant for the field of TFEB and neurodegeneration research, more evidence supporting the authors' hypothesis is required before their story is completely sound. Mainly, the authors' elimination of macroautophagy as responsible for TFEB-reduction of phospho-tau is incomplete and needs further data to support this conclusion. There is a chronic lack of western blot densitometry that needs to be remedied in order to properly quantify observed effects.

Response: We appreciate the reviewer's recognition of the novelty and significance of our findings. While our microarray analysis showed that the general autophagy pathway genes were not significantly induced in TFEB-treated mice, it was not our intention to exclude the contribution of macroautophagy in TFEB-reduction of phospho-tau. In fact, our data that PTEN is a direct target of TFEB implicates a role of PTEN-mediated autophagy in pTau clearance. However, we agree that data presented in the original submission was limited and indirect. We have performed autophagy-related experiments as suggested by the reviewer, as detailed in specific response to Figure 5. The lack of western blot densitometry is a concern shared by other reviewers and is motivated in part by the clear differences seen in most cases. We apologize for this oversight and have provided adequate quantifications. The details of which can be found in responses under specific concerns.

One issue that must be addressed is that the authors' final statement that 'TFEB was effective when introduced both before and after the onset of neuropathology' is incorrect. While they do indeed report both pathology and phenotype rescue in their P0 injected Tau mice, only Tau staining is shown to be reduced in 2M injected mice. No behavior rescue is reported for after-onset treatment. Thus, care should be taken in making such broad statements regarding therapy development.

Response: This is an excellent point and we thank the reviewer for bringing it up. We agree that we only have limited data on the adult injection experiment. We did not perform behavioral analysis due to the limited transduction efficiency by AAV when introduced to adult brains. This is an important question from both mechanism and therapy point of view and we are using genetic approaches to better address this question. Therefore, although our preliminary results are encouraging, more analysis is needed we have removed this claim in the revised manuscript.

Specific concerns:

Figure 1:

1A) The authors do not report expression pattern or levels of TFEB transduction after injection in P0 mice. Whereas they show very nice coverage with their AAV-GFP control vector (Suppl Figure 1), no TFEB staining is shown. This confounds their results, as it is impossible to determine expression levels in relevant brain regions or cell-to-cell analysis for decreases in tau.

1C) No western blot of TFEB expression is shown, which is necessary to compare TFEB expression levels between Wt and Tau mice. This is particularly important as the qRT-PCR for TFEB expression levels shown in Suppl Figure 1 does seem to indicate differential expression of TFEB in both lines of mice. Since such data is shown for 2 month-old injected mice, it is unclear why this was not analyzed for P0 mice as well. Furthermore, authors state that there are no differences in total

tau protein levels in Tau mice after TFEB injection. However, western blot shown is overexposed and non-quantifiable. Furthermore, variability of triplicates shown on the western blot in 2C does not reflect the tight error bars shown in the quantification graph in 2D, particularly for AT8 and MC1 probing.

1F) Quantification of TFEB reduction of insoluble Tau levels is missing.

Response: We have tried 3 TFEB antibodies (from Cell Signaling, Abcam and Abmart respectively), none works well on immunostaining, making it difficult to convincingly document the differences between injected vs. endogenous TFEB. We have therefore performed the following experiments to address the reviewer's questions:

1A) To assess the cell-to-cell expression, we performed and compared AAV-GFP, AAV-TFEB injections or AAV-GFP/TFEB coinjections in P0 Wt and Tau mice. The results showed similar GFP staining patterns and effects in AAV-GFP and AAV-GFP/TFEB injected Tau mice (Fig. 3A). Based on these observations, we used GFP signals to monitor the TFEB expression and correlate with pTau staining in AAV-GFP/TFEB injected Tau mice and control AAV-GFP injected Tau mice. We detected abundant GFP/pTau double positive neurons in AAV-GFP controls but minimal double positive cells in AAV-GFP/TFEB injected brains (Fig. 3A). The same effect is seen in the DOX-inducible cell line (Fig. 3B). These results suggest that TFEB expression is positively correlated with reduced pTau;

1C) We have provided western blots of TFEB in Wt and Tau mice with or without TFEB injection along with the quantifications (Fig. 1D-F and Fig. S1D&E). Overall mRNA and protein levels were comparable (Fig. S1C&E). The TFEB protein overexpression in Wt mice was slightly higher as compared to the Tau mice (Fig. 1D). Although the reasons for the differences are not clear, the western blot result argues against the possibility that the lack of adverse effect of TFEB on Wt mice is caused by lower TFEB expression.

We have provided properly exposed and quantifiable western blots for total Tau (Fig. 1C) along with the quantifications (Figs. 1E, 1F, 2B, 2C and 2E). As to the apparent inconsistency between the western image and quantification, it was due to the fact that multiple animals and brain areas, not shown in the blots, were included in the quantification. However and as explained above in the overall response, we have deleted the adult injection experiment in the revised manuscript.

1F) Quantification of TFEB reduction on insoluble Tau is now provided (Fig. 2C).

Figure 2.

2A) No GFP channel is shown for control injected mice, which makes it difficult to compare reduced Tau staining in TFEB-only expressing cells.

2B) Again, total Tau and tubulin western blots shown are overexposed.

Response: We have provided GFP images in control and TFEB injected Tau mice, showing mutually exclusive nature of GFP/TFEB with pTau (Fig. 3A). As explained above in the overall response, we have deleted the adult injection experiment in the revised manuscript and removed the associated overexposed figure.

Figure 3.

Authors claim that their effect is cell-autonomous based on inducible cell-line results shown in Figure 2. However, in this in vivo pathology rescue experiment, they do not show TFEB staining channel to confirm that surviving NeuN+ positive cells are indeed also TFEB+. This is particularly important as we also see reduced astroglial/inflammatory response in Figure 3C-F, suggesting a non-autonomous astrocyte-mediated response could also underlie improved neuronal survival. Quantification of cell number, particularly in the affected CA1 area of the hippocampus, with and without TFEB would also be desirable. Also, is neuronal survival improved in 2M old injections? Brain weight is not a direct measure of neuronal survival, so 3F needs to be supported by similar analysis as done in P0 injected mice in 3A.

Response: We performed cell culture experiment to support our in vivo finding that GFP/TFEB-positive cells are pTau-negative (Fig. 3A). We felt that TFEB/NeuN double staining will not be informative because not all neurons die without exogenous TFEB expression, as evidenced by NeuN+ neurons in non-treated (Fig. 4A, Tau) or GFP-treated Tau transgenic mice (Fig. S1B). With regards to reduced astrogliosis and microgliosis, these can be expected when general neuronal health is improved. Nevertheless, as the reviewer correctly pointed out, it is possible that non-autonomous astrocyte-mediated response could contribute to improved neuronal survival. This is an important question and we are addressing it using genetic approaches. Although this is beyond the scope of current manuscript, we have included the following statement in the Discussion (p. 12):

“Our results that TFEB and pTau stainings are mutually exclusive support a neuronal-intrinsic role of TFEB in pTau reduction. However, it remains possible that other cell types, such as astrocytes and microglia, may also participate in TFEB-dependent pTau clearance in a non cell-autonomous manner. In this regard, it is interesting to note that TFEB has been reported to mediate not only lysosomal biogenesis but also lysosomal exocytosis (Medina et al, 2011). It is difficult to probe the cellular mechanisms using the current system as the AAV vector infects both neurons and non-neuronal cells. Genetic expression of TFEB in these distinct cell types is better suited to address this question.”

As the reviewer suggested, we have performed unbiased stereology on P0-injected mice and the data is provided (Fig. 4C). The result is consistent with the NeuN staining. We have attempted to evaluate the neuronal survival in 2M-injected mice. Unfortunately the data was not conclusive due to the localized expression by stereotaxic injection at this age. This is no longer relevant as we have removed experiments related to 2M old injection.

Figure 4.

Synaptic alterations in these Tau mice have already been reported (Kopeikina et al 2013). Authors do not show comparison between Wt and Wt + TFEB in their microarray analysis in 4E.

Response: The Kopeikina et al. reference has been added (p.8). We did not report the Wt and Wt + TFEB microarray data because we did not find significant changes of synaptic pathway genes. We have added the following statement in the revised manuscript (p.8): “The fact that we did not find significant changes of the synaptic pathway genes by comparing wild-type mice with or without TFEB injection (not shown) supports the notion that the increased synaptic gene expression in TFEB-treated Tau mice is due to the rescue of Tau-triggered synaptic protein reduction rather than a dominant effect of TFEB.”

Figure 5.

Based on their microarray analysis, the authors propose that, in vivo, TFEB upregulation leads to a coordinated early up-regulation of autophagy followed by a chronic up-regulation of lysosomal-mediated clearance. While this is particularly interesting for the field of TFEB and autophagy research, and deserves to be expanded more, the functional characterization of this possible pathway is not explored properly. Only readout of lysosomal clearance shown is increased LAMP1 protein levels on a western blot (and the effect is quite modest), and this does not reflect functional upregulation of lysosomes. Additional experiments are required for this conclusion. Furthermore, the conclusion that since there is 'no appreciable transcriptional regulation of the autophagy pathway' after TFEB injection means there is no functional autophagy induction is an over-interpretation of their results. While indeed they see no significant expression changes of autophagy genes after TFEB induction, the autophagy pathway functions mostly by activating already synthesized-proteins, via post-translational modifications or sub-cellular relocalization. Thus, in order to conclude that autophagy is indeed not responsible for decreased levels of insoluble tau, the authors need to do more exhaustive analysis of their in vivo data, such as p62 or LC3 puncta staining (and their colocalization, or lack thereof, with tau aggregates) in hippocampal neurons +/- TFEB injections. Processing and conjugation of other Atg proteins (such as Atg5-Atg12 or Beclin-1) is also an option. Finally, to determine if this effect is indeed independent of macroautophagy, blocking macroautophagy by Atg5/7 knockdown or spautin treatment should also be considered.

Response: We completely agree that autophagy activity can be regulated at multiple levels, particularly via post-translational modifications. We performed transcriptional profiling because many of the autophagy genes are shown to be TFEB targets (Settembre et al., Science, 2011). However, it was not our intention to use the absence of transcriptional upregulation of general autophagy genes as evidence to exclude the contribution of macroautophagy in TFEB-reduction of phospho-tau. In fact, a role of autophagy in Tau degradation has been reported by several groups (Wang et al., Autophagy, 2010; Kruger et al., Neurobiol Aging, 2012; Caccamo et al., Aging Cell, 2013; Schaeffer et al., Brain, 2012). Our data that TFEB induces LC3-II (revised Fig. 6B) supports a role of autophagy in TFEB reduction of pTau. We agree with the reviewer's assessment that more data is needed to determine a functional contribution of autophagy in TFEB clearance of pTau. To this end, we have attempted to perform LC3/pTau staining in vivo and found it difficult to detect LC3 puncta in hippocampal neurons regardless of TFEB state, likely due to their small cytoplasm. This view is shared by speaking with autophagy experts (Xuejun Jiang and Zhenyu Ye). The use of p62 as a readout for autophagy in the context of TFEB expression is complicated because p62 is also a TFEB target.

We have performed a series of in vitro experiments related to the autophagy and lysosomal pathway as seen in the significantly expanded Fig. 6. In particular, we show that PHF1-positive staining can be detected within autophagosomes recognized by the double membrane LC3 puncta and that cells with significant LC3 puncta, indicating increased autophagy, had dramatically lower PHF1 levels compared to nearby LC3 puncta-negative cells (Fig. 6C). These results support the view that autophagy activation is associated with pTau degradation. The reviewer was concerned about the limited lysosome data presented. In particular, we only showed Lamp1 western blot without demonstrating a functional role of lysosome in TFEB reduction of pTau. In the revised version, we have provided additional data to show that TFEB expression is correlated with higher Lamp1 and lower pTau on a cell-to-cell basis (Figs. 6D and 6E) and that blocking the lysosomal activity with Leupeptin led to increased pTau and total Tau levels (Figs. 6F and 6G). We would like to point out that the effect appears to be rather modest. This is due to the highly sensitive feedback regulation of TFEB activity in response to lysosomal blockage (Settembre et al., EMBO J., 2012).

We have also performed autophagy blockage experiment by treating the Tau-inducible cells with spautin (Sptn) as the reviewer suggested. Although spautin indeed inhibited autophagy, as documented by lower LC3-II levels, we consistently observed reduced rather than elevated pTau and total Tau levels (see figure below). We found out that this is because spautin treatment resulted in not only higher TFEB levels, but also a shift to unphosphorylated and active form. The

mechanism for this intriguing spautin-induced TFEB activation is not clear and we are actively pursuing this. However, it is beyond the scope of the current investigation. We provide this data as confidential and for reviewers only information. {Figure removed upon authors' request}

Based on the new results and incorporating the review's comments, we have included the following discussion concerning the role of ALP in TFEB-mediated pTau reduction (p. 12-13): "Further studies are needed to delineate the contribution of autophagy-dependent vs. -independent lysosomal clearance of pTau and to what extent PTEN-mediates ALP downstream of TFEB. In addition, it is important to point out that in vitro cell cultures were used for mechanistic studies. The nature of pTau expressed in the acutely induced cells and in vivo is likely distinct, especially with regards to its aggregation status. Therefore, the relevance of the signalling pathways identified here to tauopathy in vivo requires further investigation and validation."

Figure 7.

7A) It is unclear what cells were used for this assay. Please add quantification of western blot data. While the effect of TFEB overexpression on AKT inhibition seems to be quite significant, the 'associated reduced pP70SK6 and pULK1' are very modest and require densitometry analysis. Overexposure of bands is also an issue in this figure panel.

7B) PTEN overexpression is running higher than endogenous PTEN. Is the transfected PTEN tagged with something or is this just a nonspecific band? Furthermore, authors use this figure to establish that TFEB-reduction in levels of P-Tau is independent of PTEN phosphatase activity on Tau protein. However, glaring overexposure of the unphosphorylated (Tau1) and total Tau blots makes it impossible to make this determination. Authors should re-expose and quantify these blots. Additionally, use of a PTEN-insensitive Tau or a phosphatase-dead PTEN are needed in this experiment (or in 7C) to verify that reductions in phospho-Tau levels after TFEB expression are not due to PTEN dephosphorylating tau.

Response: This part has been significantly revised and the current version is focused on the lysosomal pathway. Cell type (N2a) has been added (p. 10), as were the quantification of western blots (Figs. 7G, 8B and 8E). Overexposure issue has been corrected in the revised Figure 8A and 8D.

The PTEN construct is not tagged. The apparent higher molecular weight is likely due to the gross overexpression compared to the endogenous PTEN and this is no longer the case with the new western blots with overexposure corrected (Fig. 8A). We have also performed the experiments using two PTEN mutants: the phosphatase-dead (C124S) and lipid-phosphatase-dead (G129E), both were shown to be deficient in reducing pTau (Fig. 8 A-C). Thus lipid-phosphatase activity of PTEN is required for mediating the pTau clearance.

Referee #2:

(Comments on Novelty/Model System):

While it is clear that TFEB over-expression has an effect in the model of tauopathy that was studied, the lack of information concerning levels and localization of the over-expressed protein is a

significant concern both for interpretation of the data and for ensuring that other labs could understand the model well enough to replicate such findings.

With respect to the effect of TFEB over-expression on amyloid plaque abundance in the 5xFAD mouse model, the conclusion is that there is no effect on plaque density. However, the representative images that were provided raised concerns about the staining and identification of the plaques. Addressing these and other concerns that I have outlined below would greatly improve the technical quality of the manuscript.

The lack of a clearly defined mechanism for how TFEB over-expression reduces the abundance of neurofibrillary tangles reduced evaluation of the medical impact. I recognize though that this would be a difficult issue to address.

Response: We agree with the reviewer's concern regarding levels and localization of AAV-TFEB expression. In the revised manuscript, we have provided western blots of TFEB in Wt and Tau mice with or without TFEB injection along with the quantifications (Fig. 1D). To assess the cell-to-cell expression, we have performed AAV-GFP, AAV-TFEB injections or AAV-GFP/TFEB coinjections in P0 Wt and Tau mice. We detected abundant GFP/pTau double positive neurons in AAV-GFP controls but minimal double positive cells in AAV-GFP/TFEB injected brains (Fig. 3A), supporting a cell autonomous effect of TFEB in pTau reduction.

As to the effect of TFEB over-expression on amyloid plaque abundance, we have performed additional experiments as the reviewer suggested and explained our results, the details of which can be found under responses to Specific Concerns #3.

With respect to the mechanistic studies, we have performed additional experiments to strengthen a functional role of the lysosome in TFEB reduction of pTau (see extensively revised Figure 6). However and as the reviewer correctly pointed out, the in vivo mechanism on how TFEB over-expression reduces the abundance of neurofibrillary tangles remain to be defined. We appreciate the reviewer's understanding that it is a difficult issue in the context of the current study. This is nevertheless an important question and we are using genetic approaches to systematically investigate the mechanisms in vivo. For the current manuscript, we have included the following discussion concerning the mechanism of TFEB-mediated pTau reduction (p. 12-13): "Further studies are needed to delineate the contribution of autophagy-dependent vs. -independent lysosomal clearance of pTau and to what extent PTEN-mediate ALP downstream of TFEB. In addition, it is important to point out that in vitro cell cultures were used for mechanistic studies. The nature of pTau expressed in the acutely induced cells and in vivo is likely distinct, especially with regards to its aggregation status. Therefore, the relevance of the signalling pathways identified here to tauopathy in vivo requires further investigation and validation."

(Remarks):

This manuscript investigates the effects of over-expression of transcription factor EB (TFEB) on two distinct mouse models of Alzheimer's disease pathology that exhibit amyloid plaques and neurofibrillary tangles respectively. The major findings reported by the authors are that TFEB over-expression reduces neurofibrillary tangle pathology in a manner that appears to depend on the TFEB-dependent up-regulation of PTEN expression. While the effects of TFEB over-expression on reducing tauopathy are impressive, the mechanisms whereby such effects are achieved, including the relevant targets downstream of PTEN, remain unclear. Elucidating such mechanisms in detail is likely beyond the scope of this manuscript. Therefore, the concerns presented below focus specifically on interpretation of the effects of TFEB over-expression on disease pathology.

Specific Concerns:

1. The authors characterize their TFEB over-expression strategy as "mild Adeno-Associated Virus (AAV)-mediated TFEB expression". However, no data is provided that compares the endogenous to the over-expressed TFEB protein levels. A 2-3 fold change was seen at mRNA level (Fig. S1). Was this for whole brain? Based on GFP expression pattern in Fig S1, the effect would likely be much greater in the cerebral cortex. As the findings of the paper depend so heavily on this TFEB over-expression strategy, a better characterization of the TFEB protein levels in relevant brain regions of control versus TFEB over-expressing mice is critical. Such a characterization should include measurement of TFEB protein levels as well as an assessment of the fraction of neurons that over-express TFEB in the relevant brain regions.

Response: Forebrain samples were used for qRT-PCR and biochemical analysis. This information has been provided in the Methods (p. 17). We completely agree with the reviewer's view that it is important to better characterize TFEB expression, and this is a point also shared by other referees. To this end, we have provided western blots of TFEB in Wt and Tau mice with or without TFEB injection along with the quantifications (Fig. 1D-F, S1D&E). We could not directly assess the fraction of neurons that over-express TFEB in the relevant brain regions as the current available TFEB antibodies do not work well on immunostaining. To circumvent the issue, we performed and compared AAV-GFP and AAV-GFP/TFEB coinjections in P0 Wt and Tau mice. The results showed similar GFP staining patterns in AAV-GFP and AAV-GFP/TFEB injected mice (Fig. 3A). As such, we believe that GFP signals could be used to correlate with TFEB expression. Quantification of GFP/NeuN double positive cells in cortex and hippocampus revealed that approximately 50% of the neurons were GFP positive. This result has been included in Fig. S1B. Taking these analyses into consideration, we realize that the term "mild" may not accurately reflect the level or activity of TFEB expressed on a per neuron basis. We have therefore avoided the use of "mild" in the revised manuscript.

2. Fig. S2. It is concluded based on the data presented in Fig. S2 that TFEB over-expression does not affect amyloid plaque abundance in the 5xFAD mouse model of Alzheimer's disease. However, the punctate staining in the images of APP antibody stained brain sections does not match the expected appearance of amyloid plaque staining in this mouse line as there are no plaques evident in the hippocampi or in superficial layers of the cerebral cortices. Thus, it is not certain that the staining in these images actually represents plaques. Analysis of staining with an additional marker of plaques such as thioflavin S is required to ensure that amyloid plaques have been properly identified in these experiments.

Response: The overall Ab staining pattern we observed in the 5xFAD mice is consistent with that of Oakley et al. (J. Neurosci, 2006) in that amyloid deposition "reaches a very large burden, especially in subiculum and deep cortical layers." Although not evident from the low-resolution images, examination of different sections did reveal positive staining in hippocampus. As requested by the reviewer, we have performed Thioflavin S staining (Fig. S2E). It is clear from the overlay of 6E10 (Amyloid) and Thioflavin S images that the staining is specific.

3. No evidence is provided to demonstrate that robust TFEB over-expression (at the protein level) was successfully achieved in the 5xFAD mice. This limits the confidence concerning the interpretation of the negative result relating to plaque abundance.

Response: TFEB protein levels have been provided (Fig. S2A). We would like to state that although we did not observe a significant difference in Ab pathology as a function of TFEB expression, we

could not conclude that TFEB does not modulate Ab. Instead our claim is that “the potent clearance of pTau/NFTs using the same TFEB injection scheme argues that Ab and pTau/NFT pathologies are subject to distinct TFEB regulations.” (p. 11).

4. In figure 1C and D, the representative blots show considerable variability yet the accompanying graphs indicate practically none. This discrepancy should be explained.

Response: This is also raised by Referee #1. The apparent discrepancy was due to the fact that multiple animals and brain areas, not shown in the blots, were included in the quantification. However and as part of the response to Reviewer 1’s overall concerns, we have deleted this figure in the revised manuscript.

5. Figure 2 appears to compare TFEB/GFP AAV injected brains to untreated controls. The ideal control would have been injection of a comparable titer of GFP AAV. A justification for this less than ideal control should be provided.

Response: This is an excellent point. We did not include GFP AAV injected mice as controls was mainly due to the fact that we have used the virus to set up injection parameters and we did not find any adverse effect with this control virus. In addition, since the injections were done at P0 before genotyping and since the frequency of obtaining Tau transgenic mice is 1/4, it is not particularly feasible to perform parallel injections of GFP or TFEB. Nevertheless, we have performed new experiments including comparable titer of GFP AAV-injected brains as controls for pTau analysis (Figure 3A) and stereological analysis (Fig. 4C).

6. For the experiments in Figure 2, is this a TFEB-GFP fusion protein or the same untagged TFEB construct that was used elsewhere?

Response: It is the same untagged TFEB coinjected with AAV-GFP. We have made it clear in the revised manuscript (p. 6).

7. What was the level of TFEB overexpression (protein levels) in the T40PL cell line that was used to achieve the effects shown in Figure 2D?

Response: As expected from transient transfection, the level of TFEB overexpression is high. Comparison with endogenous TFEB expressed in the T40PL cell line may not be very meaningful as the latter is exceedingly low (Fig. 8D). Our rough estimate is >30-fold.

8. Over-expression of TFEB (and closely related transcription factors) causes renal carcinoma in humans (PMID: 21670463). This link between TFEB and cancer raises concerns about the use of therapeutic strategies that would intentionally raise the expression of this protein. As the findings of the current manuscript have the potential to raise the hopes and expectations of patients and/or the families of patients who suffer from Alzheimer's disease, it is important that such caveats are also acknowledged.

Response: This is an important point indeed. We have included the reference as well as the

following cautionary notes in the Discussion (p. 14): “However, the current work represents a proof-of-concept study. A TFEB-based therapy likely requires the identification of specific small molecule TFEB activators. Furthermore, it is important to note that, as a master regulator of lysosomal activity, expression levels and duration of TFEB need to be properly controlled. Indeed, dysregulation of members of the microphthalmia family of transcription factors including TFEB have been shown to cause renal carcinomas (Haq & Fisher, 2011). Therefore, rigorous studies are needed to evaluate the safety profiles of potential TFEB activators.”

Referee #3.

(Remarks):

This is an interesting and potentially significant set of findings on the ability of AAV-mediated TFEB expression to prevent or rescue diverse aspects of disease in a mouse model of tauopathy. The data demonstrate that PTEN is a direct target of TFEB, which is required for TFEB dependent tau clearance. The authors persuasively demonstrate that TFEB expression reduces levels of phosphorylated and misfolded tau species. They conclude that the mechanism involves a selective targeting of these altered tau species in both soluble and insoluble form to the autophagy pathway where they are degraded within lysosomes. The selective targeting is somewhat unanticipated given previous studies indicating that tau aggregates are a selective target for autophagy and other studies suggesting that all forms of tau, especially when overexpressed, may be substrates for autophagy. The major concern in the study is that, while it is possible that pathogenic forms of tau are selectively targeted by TFEB mediated autophagy, the evidence that the observed changes on tau are in fact autophagy and lysosomal dependent is not firmly supported by the data presented. In addition, there is no clear mechanism to account for the selectivity either demonstrated experimentally or discussed that would gain support from the known effects of TFEB demonstrated in this system. These issues could be addressed experimentally as discussed below.

1. The evidence that TFEB is altering tau through an autophagy/lysosomal-dependent mechanism is mainly based on the previous work by these authors, and evidence presented in the report establishing a lysosomal-dependent mechanism for tau changes is circumstantial. The strongest case, which is still indirect, is made by the cell studies in which total tau levels are decreased in both soluble and insoluble fractions. Further studies are needed to demonstrate lysosomal dependence, such as analyses involving the clamping of lysosomal proteolysis with an inhibitor and/or demonstrating the presence of the tau substrates in lysosomes and ideally also directly demonstrating that lysosomal activity is enhanced as predicted. It is not clear from the data presentation whether the mechanism proposed is solely related to an enhancement of lysosomal activity rather than, or in addition to, a change in autophagy induction. The discrimination of these two possible mechanisms is important since it is not clear how the authors would explain the selective elimination of phosphotau species by only an enhancement of lysosomal activity, except possibly through CMA.

Response: This concern is shared by Reviewer 1 and the referee made some excellent suggestions. We have performed a series of in vitro experiments related to the autophagy and lysosomal pathway as seen in the significantly expanded Fig. 6. In particular, we show that PHF1-positive staining can be detected within autophagosomes recognized by the double membrane LC3 puncta and that cells with significant LC3 puncta, indicating increased autophagy, had dramatically lower PHF1 levels compared to nearby LC3 puncta-negative cells (Fig. 6C). These results support the view that autophagy activation is associated with pTau degradation. We have also performed clamping experiment as suggested and the result showed that blocking the lysosomal activity with Leupeptin

led to increased pTau and total Tau levels (Fig. 6F). We would like to point out that the effect appears to be rather modest. This is due to the highly sensitive feedback regulation of TFEB activity in response to lysosomal blockage (Settembre et al., EMBO J., 2012). In addition, we show that TFEB expression is correlated with higher LAMP1 and lower pTau on a cell-to-cell basis (Fig. 6D). We believe that these studies provide direct evidence for a lysosomal-dependency of TFEB-mediated pTau clearance. To evaluate a functional contribution of autophagy, we performed an autophagy blockage experiment by treating the Tau-inducible cells with spautin. The results are intriguing in that spautin treatment resulted in higher TFEB and lower total Tau and pTau levels (see figure under response to Reviewer 1, Figure 5.) The mechanism of which is under investigation.

2. The in vivo studies are subject to the same concerns, which are heightened by the fact that total tau levels are not changing after TFEB expression. This observation is interpreted as reflecting selective degradation of phosphotau and misfolded tau species although observed tau changes could alternatively be explained by alterations of tau phosphorylation. The authors attempted to exclude the possibility of a change in phosphorylation by performing western blot analyses of relevant protein kinases and phosphatases but this analysis is minimally informative because the phosphorylation states of the protein kinases, which are the only appropriate way to assess kinase activation changes using immunoblot analysis, were not monitored. In the case of cdk5, p25 levels need to be measured to assess activation. The panel of possible tau phosphatases analyzed is also incomplete. Based on the collective data in this report, therefore, the possibility that these changes relate to effects of TFEB expression on tau phosphorylation cannot be excluded. Shifts in phosphorylation could well account for differential effects on distributions of phospho and total tau species in the soluble and insoluble pools. In the absence of direct evidence for a lysosomal-dependent pathway and a mechanism for selective targeting for phosphotau species to lysosomes, the possible effects on phosphatases or kinases has to be excluded rigorously.

Response: We completely agree with the reviewer's point that western blot analyses of kinases and phosphatases are minimally informative. Accordingly we have removed the western blots from the supplemental information. We feel that it is not possible to exclude the effects of all phosphatases or kinases as there are too many potential candidates. To address the possible effects of TFEB on tau phosphorylation/dephosphorylation, we have performed rigorous quantifications of different Tau species in detergent extractable brain lysates (Fig. 1F), soluble and insoluble fractions of the brain lysates (Figs. 2B and 2C), and the Tau inducible cell line (Fig. 2E). Our results show that, in all the preparations, while the AT8 (CP13)-positive pTau were reduced, the corresponding unphosphorylated Tau recognized by Tau1 remained constant. This result argues against a shift in phosphorylation. Further, we have performed additional experiments to gain "direct evidence for a lysosomal-dependent pathway" in TFEB-mediated pTau clearance as detailed in the above responses. Collectively we believe that we have provided strong evidence that TFEB targets phosphorylated and misfolded Tau species for degradation. However, as the reviewer stated, we cannot exclude a possible role of dephosphorylation in TFEB reduction of pTau and we have included this statement in the revised manuscript (p. 11): "although it is clear that TFEB targets the detergent-insoluble pTau for degradation, we cannot exclude the possibility that TFEB may affect Tau phosphorylation/dephosphorylation at Tau1-independent sites in the soluble pool."

3. There are several concerns about the immunoblot analyses of tau fractions which further hinder the evaluation and interpretation of these results. There is the general concern that none of these data are quantified and some involve very small numbers of animals. Information is not evident about how much of each tau fraction was loaded onto gels or whether immunoblots of different fractions can be directly compared (i.e., run on the same gel or exposed for equal times). This results in what may be a misleading impression that insoluble and soluble tau are equally abundant. Without information about the relative abundance of these tau forms, it is difficult to interpret, for example, how a change in insoluble tau forms may or may not impact the soluble pool and whether changes in insoluble forms reflect proteolytic loss rather than disaggregation and redistribution to a

soluble pool. The choice of gamma tubulin as a loading control, rather than a more conventional tubulin species or another protein, is unclear and the result presented gives the impression that insoluble and soluble gamma tubulin are equally abundant, which would be surprising.

Response: This is a concern shared by other reviewers and we again apologize for not performing adequate quantifications. In this revised version, we have performed all necessary quantification and provided the loading information as requested by the reviewers. As for the sample number, we realize that we were not precise in our reporting as most of them involve multiple independent experiments and the sample numbers reported represent only one analysis in certain cases. This is because the breeding only generates ¼ of each wild-type and Tau transgenic mice. We have made the sample number description more explicit in the current version as stated in the figure legends. In addition we have performed additional in vivo and in vitro experiments and the sample numbers have been increased accordingly.

We have also added more details on our brain lysate fractionation procedure in Material and Methods section, so that readers would have a better idea about the amount of proteins loaded on to each gel. In short, 1/140 of cytosol fractions and 1/10 of insoluble fractions were loaded per western blot analysis. Tau levels in soluble and insoluble fractions are not to be compared directly for the absolute amount, but more for the purpose of comparing TFEB's effect within each fraction. Since it is difficult to measure the protein concentration in the insoluble fraction, even if we ran both fractions on the same gel to receive the same exposure, it still could be misleading regarding the exact amount of different species of tau is in each fraction. We felt that it is better that we separate the two, so that we can find the best exposure for each fraction to represent the main findings here, that is the effect of TFEB can be observed for both soluble and insoluble tau species. Following the suggestion from the reviewer, we included the ratio of the protein preparation loaded per gel, in order to avoid the mis-impression of the amount of gamma tubulin in each fraction.

As for the loading control, gamma tubulin is ubiquitously expressed in eukaryotic cells and thus, is also routinely used as loading control, as are actin, GAPDH, other forms of tubulin, and several other proteins. Many high quality research publications can be found using gamma-tubulin as loading control without providing explanation. Our lab normally uses gamma-tubulin, beta-actin, or GAPDH as loading control. We haven't observed any differences among these three in the assays we have done. Therefore we don't think the choice of using gamma tubulin here will affect the conclusion of our results.

4. If it could be established firmly that ptau is selectively targeted to an autophagy system, this finding and the functional rescue of the animals by TFEB, are sufficiently exciting with or without a clarification of the targeting mechanism, but it would be helpful to have a more rigorous discussion of the possible mechanisms for selective targeting of ptau, including how a CMA mechanism might operate and also be consistent with the current data.

Response: We appreciate that the referee finds our study exciting. We agree that more studies are needed to define the targeting mechanism and, as a follow up analysis, we are using genetic approaches to systematically investigate the TFEB function. As suggested by the reviewer, we have revised our manuscript to provide more rigorous discussion of possible mechanisms as follows (p. 12-13): "Further studies are needed to delineate the contribution of autophagy-dependent vs. – independent lysosomal clearance of pTau and to what extent PTEN-mediates ALP downstream of TFEB. In addition, it is important to point out that in vitro cell cultures were used for mechanistic studies. The nature of pTau expressed in the acutely induced cells and in vivo is likely distinct, especially with regards to its aggregation status. Therefore, the relevance of the signalling pathways identified here to tauopathy in vivo requires further investigation and validation. The mechanism mediating the macroautophagy-independent lysosomal degradation of pTau remains to be investigated. One possibility is through chaperone-mediated autophagy (CMA). In this regard, Tau has been reported to contain two CMA targeting motifs obligatory for hsc70 binding and LAMP2A-mediated lysosomal degradation (Wang et al, 2009). However, only proteolytically processed Tau

fragments were shown to be subject to the CMA pathway (Wang et al, 2009). Since we did not detect appreciable levels of Tau fragments, the role of CMA in our system is not clear.”

Additional comments/questions

1. For at least some of the tau immunoblot data, a presentation of the entire length of the gel should be included to allow evaluation of additional tau species, including breakdown products, which would be expected to be present and potentially informative.

Response: Examples of full-length tau immunoblots are provided below as requested:

All these are from films with the longest exposure. The antibodies we used were very specific and we normally do not see other less abundant species of tau using these antibodies. The strip on the side shows the protein marker used for these gels.

2. It is not clear which supernatant fraction is loaded in Figure 1E.

Response: The fraction after the first ultracentrifugation is considered as cytosol fraction as stated in the methods section. And this fraction was used as detergent-free soluble fraction.

3. The tau Dako antibody, used in this study as a measure of total tau, is stated in the Dako website to be recognizing non-phosphorylating tau.

Response: The Dako website states that “the antibody labels tau protein independently of phosphorylation state” (–A0024 product insert) which means that it recognizes tau with or without phosphorylation and, accordingly, total tau.

4. *Ages of APP mice analyzed should be provided.*

Response: The age of APP mice analyzed (4 months) have been provided in Figure S2 legend.

Once again we thank the reviewers for their constructive comments. We believe that, by incorporating the reviewers' suggestions, our manuscript is vastly improved, and we hope that the referees are in agreement with us.

2nd Editorial Decision

13 May 2014

Thank you for the re-submission of your manuscript to EMBO Molecular Medicine. We have now heard back from the three referees whom we asked to re-evaluate your manuscript.

As you will see, despite still some suggestions to improve the study (both in terms of experiments and rewriting here and there), referees 1 and 3 are now globally supportive of publications. However, referee 2 is still very much concerned about conclusions being not fully supported by the data and provide a list of items to check and modify, again by performing additional experiments but also rephrasing to be more cautious in drawing conclusions. We feel that these comments are important and need to be carefully dealt with. We are willing to give you another chance to address all these remaining issues but with the understanding that all three referees will have to be fully satisfied for the manuscript to be accepted and as such I would strongly suggest to follow all the suggestions.

I look forward to seeing a revised form of your manuscript as soon as possible.

***** Reviewer's comments *****

Referee #1 (Remarks):

In this re-submission, Polito & Li et al have addressed most of the concerns with the original manuscript. First, they have properly quantified their results, which allows for much more direct analysis and comparison of their results. Second, they have attempted to detect transduced TFEB in vivo and while their analysis remains indirect due to technical issues, the authors do show correlative data that is satisfactory. Third, they have strengthened their mechanistic analysis of PTEN as necessary for TFEB mediated pTau clearance using phosphatase dead PTEN. Given these improvements, and the novelty and significance of their findings for the field of neurodegeneration, the revision is encouraging.

Remaining points that must be corrected before publication:

Figure 3: the finding that TFEB+ cells are consistently tau negative is quite interesting, and suggests efficient clearance of tau by TFEB. However, to eliminate possible artifacts of staining/transfection, a time course depicting initial co-expression of both tau and TFEB in cells and a progressive reduction in tau staining is necessary. With the doxycycline-inducible system, this should be fairly

easy to accomplish.

Figure 6: Panel C: The use of EGFP-RFP-LC3 as a reporter for autophagy puncta determination seems somewhat contrived, particularly since no autophagy flux determination (ratios of yellow to red vesicle formation) was performed. Furthermore, the images shown seem to reflect a paucity of LC3 puncta, suggestive of little to no autophagy occurrence in these cells, even in those that remain PHF1 positive. Furthermore, the figure panel shows most (if not all) of the puncta to be both present in the red and green channels. This would indicate autophagy flux blockage, rather than autophagy induction. Since Figure 6F (and its quantification graph 6G) nicely shows LC3II levels increasing beyond control cells after leupeptin treatment in TFEB transfected cells, this is clearly not the case. Better quality images, and proper quantification of vesicle numbers (ALs vs. APs vs total) should clarify the situation.

Referee #2 (Comments on Novelty/Model System):

The effect of TFEB over-expression of Tau metabolism represents an interesting phenomenon. The experiments that were performed appear to be of good technical quality. The combined use of in vivo and in vitro models is sensible. Unfortunately, the conclusions that the authors reach are over-reaching and are not unambiguously substantiated by the data.

Referee #2 (Remarks):

The TFEB-dependent effects (Degradation? Altered phosphorylation?) represents an interesting phenomenon that could have relevance for Alzheimer's disease and is likely to be of interest to many people in this field. A major weakness of the paper is that there is limited mechanistic insight into how such regulation is achieved. For example, a role for PTEN is implied from in vitro studies but was not confirmed in the in vivo model. There is also an over-selling of conclusive statements that favor their chosen model when in fact the actual data is compatible with multiple interpretations. Specific concerns are outlined below.

1. The negative data regarding lack of effects of TFEB over-expression in the AD model are hard to interpret. Does this mean that TFEB over-expression in this model does not influence lysosome function or rather that lysosome function is not relevant to Abeta metabolism (a conclusion that stands in sharp contrast to many other studies on this topic). The 5xFAD mouse strain that is repeatedly referred to as "APP mice" actually transgenic for mutant forms of both APP and presenilin 1 (PS1). There is now extensive evidence that PS1 is important for lysosome function. There are so many variables at work here that the negative is very difficult to interpret. Thus, rather than broadly concluding that TFEB over-expression does not affect Abeta, it should be more narrowly emphasized that this particular over-expression and analysis strategy at a particular time point in this one particular model of Alzheimer's disease did not yield a positive result without absolutely ruling out a role for TFEB and/or lysosome function in Abeta metabolism in other mouse models or more importantly in the human disease.

2. The following conclusion from the data in Figure 1 is overly definitive: "since Tau1 levels remained constant, reduction of AT8-positive Tau indicates that TFEB expression leads to the degradation rather than dephosphorylation of the phospho-Tau (pTau)." If the phosphorylated pool of Tau were a small fraction of the total, it would be possible to achieve large reductions in the pTau pool without substantially increasing the pool detected by the Tau1 antibody. Alternatively, given the long time course of the experiment, how is it shown that there is not simply a lower rate of Tau

phosphorylation as opposed to enhanced clearance? Thus, this conclusion should be clearly presented as speculation.

3. With respect to Figure 2, the authors conclude that the reduction in levels of detergent insoluble pTau when TFEB is over-expressed in vivo "demonstrates" a role for TFEB in targeting pTau for degradation. However, it is not possible to make such definitive claims based on the steady state levels that are observed after months of altered TFEB expression. An equally plausible alternative explanation is that there is simply a reduced rate of Tau phosphorylation under such conditions. In fact, the authors do acknowledge this possibility later on in discussing the results from their in vitro studies but nonetheless strongly favor the degradation model rather than giving equal weight to both possibilities until further data can resolve the issue.

4. Figure 6B. It is not correct to state that "TFEB could indeed induce LC3-II expression". As this change in LC3 electrophoretic mobility reflects a post-translational modification rather than a gene expression change. Furthermore, as the effect was only very transient (24-48 hours post-transfection) it is not clear that it is relevant for explaining the long term changes in the brain following months of TFEB over-expression. If the goal is to show that TFEB-dependent regulation of autophagy is the relevant mechanism for clearing pTau from the brain, then the measurements should be made in the brain.

5. The only evidence to support a change in lysosome abundance/function in the brains that over-express TFEB appears to be the picture of LAMP1 staining in 1 cell with nuclear TFEB in panel 6D. If the major message of the paper is to be that TFEB clears pTau in vivo via a lysosome-dependent mechanism, there needs to be a more robust demonstration that lysosome abundance/function is actually increased in this model. The mRNA studies suggest changes that are consistent with this model but are ultimately meaningless if not robustly corroborated at the protein/organelle level.

6. It is proposed based on transient transfection in cell culture studies that TFEB-dependent up-regulation of PTEN is critical for mediating TFEB-dependent pTau clearance. However, this major point of the study was not confirmed in vivo? Is PTEN up-regulated in the brain in response to TFEB over-expression? Does the degree of PTEN over-expression correlate at the cellular level with pTau clearance? While the results of such experiments would at best be correlative, they would constitute the minimum required to consider the idea that this one out of all the hundreds of TFEB-regulated proteins could have be of such high relevance for the phenomenon under investigation.

7. Figure 7 shows that TFEB over-expression promotes up-regulation of PTEN. The degree of TFEB over-expression required to yield this effect should be quantified and shown in order to help the reader assess if this is likely to be relevant for the situation of modest TFEB overexpression in vivo.

8. Was TFEB identified as a PTEN target in the author's previous studies? For example: Sardiello et al, Science, 2009; Palmieri et al, Hum Mol Genet, 2011?

9. Fig. S6. Why is the lysosomal vATPase subunit (Atp6v0d2) whose expression does not change in response to TFEB over-expression grouped with the Autophagy genes rather than the Lysosome genes? If it were grouped with the other lysosomal proteins, the message of this figure would have been greatly diluted as it would suggest that the up-regulation of lysosome genes is not uniform. This raises questions about how the 3 representative examples of lysosome genes (Ctsa, Ctsd, Mcoln1) were selected.

Referee #3 (Remarks):

As commented in the original review "If it could be established firmly that ptau is selectively targeted to an autophagy system, this finding and the functional rescue of the animals by TFEB, are sufficiently exciting with or without a clarification of the targeting mechanism.

The authors have now provided some additional support for the targeting of tau to lysosomes by showing increased PHF1 immunoreactive tau after leupeptin treatment and an example of colocalization of PHF1 immunoreactivity in an LC3-positive compartment. In conjunction with the TFEB effects, the case is reasonable for a role of autophagy in p-tau processing. Proposed explanations for the selective targeting of p-tau are now discussed and the possible involvement of CMA is raised, although the new data showing the colocalization of PHF1 immunoreactivity in an LC3-positive compartment seem to suggest a macroautophagy route. Clearly, however, the mechanism will need to be more thoroughly worked out in a follow-up study. The current data provide impetus for these further investigations.

A second earlier concern is whether the TFEB-related changes in autophagy are the sole or main basis for the reduced tauopathy, independently of induced changes in other processes including changes in phosphatases or kinases. In the absence of new data addressing this point, this remains a distinct possibility and the authors have now included statements of caution that acknowledge this possibility. Further statements in the response are, however, puzzling : "Our results show that, in all the preparations, while the AT8 (CP13)-positive pTau were reduced, the corresponding unphosphorylated Tau recognized by Tau1 remained constant. This result argues against a shift in phosphorylation" . If not reflecting a change in phosphorylation state, what other process would account for this set of findings? Also for the statement "although it is clear that TFEB targets the detergent-insoluble pTau for degradation, we cannot exclude the possibility that TFEB may affect Tau phosphorylation/dephosphorylation at Tau1-independent sites in the soluble pool" how have the authors excluded autophagy-unrelated effects on detergent-insoluble pTau in addition to any autophagy-related ones?

Other earlier concerns have been addressed well with additional data.

We would like to thank the reviewers once again for their constructive re-review of our manuscript “Selective Clearance of Aberrant Tau Proteins and Rescue of Neurotoxicity by Transcription Factor EB.” We are grateful to the editor for giving us another chance to revise our manuscript. We have since included additional data and rephrased certain statements as requested by the reviewers. Below are the reviewers’ critiques (in *italics*) followed by our point-by-point responses.

Referee #1:

In this re-submission, Polito & Li et al have addressed most of the concerns with the original manuscript. First, they have properly quantified their results, which allows for much more direct analysis and comparison of their results. Second, they have attempted to detect transduced TFEB in vivo and while their analysis remains indirect due to technical issues, the authors do show correlative data that is satisfactory. Third, they have strengthened their mechanistic analysis of PTEN as necessary for TFEB mediated pTau clearance using phosphatase dead PTEN. Given these improvements, and the novelty and significance of their findings for the field of neurodegeneration, the revision is encouraging.

Response: We appreciate the reviewer’s recognition of the novelty and the significance of our findings and their acknowledgement that our initial revision addresses most of his/her concerns.

Specific Concerns:

Figure 3: the finding that TFEB+ cells are consistently tau negative is quite interesting, and suggests efficient clearance of tau by TFEB. However, to eliminate possible artifacts of staining/transfection, a time course depicting initial co-expression of both tau and TFEB in cells

and a progressive reduction in tau staining is necessary. With the doxycycline-inducible system, this should be fairly easy to accomplish.

Response: We have performed the time course experiment in the Tau-inducible cell line by monitoring the pTau immunoreactivity every 8 hours up to 40 hours (new Figure 3B & 3C, showing only 8, 16, and 40hr time points). We took immunofluorescence images using confocal microscope with FLAG-TFEB in the green channel (Alexa488) and PHF1 staining in the deep red channel (Alexa647) to minimize bleed-through effects between fluorophores. We performed quantitative colocalization analysis using “coloc 2” plugin in Fiji (ImageJ), where threshold of the images were determined automatically to avoid subjective manipulation of the images. Eight projection slices per view field and 4 view fields per time point were used in the calculation. The results showed that TFEB intensities began to be negatively correlated with PHF1 staining starting at the 16 hour time point, while GFP intensities showed no correlation with PHF1, as presented in Pearson correlation coefficients (Pearson R values) in Figure 3C. We believe this is an insightful experiment and its inclusion at the behest of the reviewer has allowed us to obtain more mechanistic insight into how TFEB expression results in the elimination of pTau.

Figure 6 : Panel C: The use of EGFP-RFP-LC3 as a reporter for autophagy puncta determination seems somewhat contrived, particularly since no autophagy flux determination (ratios of yellow to red vesicle formation) was performed. Furthermore, the images shown seem to reflect a paucity of LC3 puncta, suggestive of little to no autophagy occurrence in these cells, even in those that remain PHF1 positive. Furthermore, the figure panel shows most (if not all) of the puncta to be both present in the red and green channels. This would indicate autophagy flux blockage, rather than autophagy induction. Since Figure 6F (and its quantification graph 6G) nicely shows LC3II levels increasing beyond control cells after leupeptin treatment in TFEB transfected cells, this is clearly not the case. Better quality images, and proper quantification of vesicle numbers (ALs vs. APs vs total) should clarify the situation.

Response: Initially we wanted to show that cells with significant LC3 puncta had lower PHF1 levels, suggesting a tight correlation between autophagy activation and pTau reduction. The pictures were displayed in low magnification view to show PHF1 staining in different cells for this purpose. However, it lost some details of LC3 punctas. To address the review's concern, we added higher magnification views in Figure 7B top panel, which showed clear red-only punctas (red arrows), indicating autolysosome formation and autophagic flux enhancement but not blockage, as a function of TFEB expression. Further, as the reviewer requested, we have performed proper image quantification on high quality images of the number of autophagosomes and autolysosomes per cell and ratio of autolysosomes to total LC3 punctas (Figure S7), and the results showed that both the number of autolysosomes per cell and the ratio of autolysosomes to total were significantly increased in TFEB overexpressing cells compared to vector-transfected controls, indicating that TFEB enhances autophagic flux.

Referee #2:

(Comments on Novelty/Model System):

The effect of TFEB over-expression of Tau metabolism represents an interesting phenomenon. The experiments that were performed appear to be of good technical quality. The combined use of in vivo and in vitro models is sensible. Unfortunately, the conclusions that the authors reach are over-reaching and are not unambiguously substantiated by the data.

(Remarks):

The TFEB-dependent effects (Degradation? Altered phosphorylation?) represents an interesting phenomenon that could have relevance for Alzheimer's disease and is likely to be of interest to many people in this field. A major weakness of the paper is that there is limited mechanistic insight into how such regulation is achieved. For example, a role for PTEN is implied from in vitro studies but was not confirmed in the in vivo model. There is also an over-selling of conclusive statements that favor their chosen model when in fact the actual data is compatible with multiple interpretations. Specific concerns are outlined below.

Response: We appreciate and respect the reviewer's critique of our revised manuscript and have made numerous changes to the language to avoid over-reaching statements in order to address their concerns regarding over-selling our finding. We now believe that the language used appropriately describes the data and properly indicates when a finding is suggestive rather than conclusive. Furthermore, in our discussion we openly examine alternative interpretations of the data and state when we are unable to outright exclude other explanations. With regard to the reviewer's comments concerning the "mechanistic insight" we agree that the in vivo mechanism with respect to how TFEB over-expression reduces the abundance of neurofibrillary tangles remain a critical question that we are actively pursuing. We however believe that the inclusion of such a mechanism is beyond the scope of this manuscript, a view originally shared by referee #2 and also shared by referee #3. We have however included additional data in the revised manuscript that provides additional potential insight into this mechanism.

Specific Concerns:

1. The negative data regarding lack of effects of TFEB over-expression in the AD model are hard to interpret. Does this mean that TFEB over-expression in this model does not influence lysosome function or rather that lysosome function is not relevant to Abeta metabolism (a conclusion that stands in sharp contrast to many other studies on this topic). The 5xFAD mouse strain that is repeatedly referred to as "APP mice" actually transgenic for mutant forms of both APP and presenilin 1 (PS1). There is now extensive evidence that PS1 is important for lysosome function. There are so many variables at work here that the negative is very difficult to interpret. Thus, rather than broadly concluding that TFEB over-expression does not affect Abeta, it should be more narrowly emphasized that this particular over-expression and analysis strategy at a particular time point in this one particular model of Alzheimer's disease did not yield a positive result without absolutely ruling out a role for TFEB and/or lysosome function in Abeta metabolism in other mouse models or more importantly in the human disease.

Response: We completely agree with the reviewer's concern and we apologize for our over generalized statement. While we acknowledge the potential of the mutant form of PS1 to be a confounding factor, the data concerning the role of PS1 in lysosome function remains controversial and, as the reviewer rightfully pointed out, there are many variables at work. To address the reviewer's concern, we have revised the manuscript as follows: a) We have changed sub-title in the Results from "TFEB targets pTau/NFT pathology without affecting A β " to "TFEB differentially targets A β and pTau/NFT pathologies" (p. 5); b) We have included the following cautionary notes in the Discussion: "It is important to note, however, this negative data should be interpreted with caution. Our finding is limited to the system we employed using the 5xFAD mice and needs to be further validated in other APP/A β mouse models and by other approaches such as genetic manipulation. It remains possible that the TFEB expression attainable in our system may not be sufficient to impact A β pathology or that APP/A β may subject to TFEB independent ALP regulation." (p.11). c) We have changed the designation of the mouse model from "APP" to "5xFAD".

2. *The following conclusion from the data in Figure 1 is overly definitive: "since Tau1 levels remained constant, reduction of AT8-positive Tau indicates that TFEB expression leads to the degradation rather than dephosphorylation of the phospho-Tau (pTau)." If the phosphorylated pool of Tau were a small fraction of the total, it would be possible to achieve large reductions in the pTau pool without substantially increasing the pool detected by the Tau1 antibody. Alternatively, given the long time course of the experiment, how is it shown that there is not simply a lower rate of Tau phosphorylation as opposed to enhanced clearance? Thus, this conclusion should be clearly presented as speculation.*

Response: Our conclusions were in-line with the other data presented in the manuscript and represent the most consistent interpretation of the results. However, we agree with the reviewer and have softened our tone the text to point out that degradation is not the only possible interpretation of the data: "Since the Tau1 levels remained constant, reduction of AT8-positive Tau most likely indicates that TFEB is involved in the degradation rather than dephosphorylation of the phospho-Tau (pTau). However and inconsistent with this view, total Tau levels were not significantly altered (Fig. 1C and quantified in 1F), suggesting that the pTau pool may represent only a small pool of total Tau levels or that TFEB might also act on Tau phosphorylation/dephosphorylation." (p. 5-6). The reviewer's comment on the possible effect of TFEB on the rate of Tau phosphorylation is excellent and we have added the following in the Discussion: "we cannot exclude the possibility that TFEB may promote pTau dephosphorylation or lower the rate of Tau phosphorylation in the soluble pool." (p. 12).

3. *With respect to Figure 2, the authors conclude that the reduction in levels of detergent insoluble pTau when TFEB is over-expressed in vivo "demonstrates" a role for TFEB in targeting pTau for degradation. However, it is not possible to make such definitive claims based on the steady state levels that are observed after months of altered TFEB expression. An equally plausible alternative explanation is that there is simply a reduced rate of Tau phosphorylation under such conditions. In fact, the authors do acknowledge this possibility later on in discussing the results from their in vitro studies but nonetheless strongly favor the degradation model rather than giving equal weight to both possibilities until further data can resolve the issue.*

Response: We thank the reviewer for pointing out this error and have changed the word 'demonstrate' to 'indicate' (p. 6) as the reviewer appropriately pointed out that the word 'demonstrate' is misleading, as we, the reviewer properly notes, already acknowledge the possibility that TFEB may affect Tau phosphorylation/dephosphorylation in the discussion section of the manuscript (see above).

4. *Figure 6B. It is not correct to state that "TFEB could indeed induce LC3-II expression ". As this change in LC3 electrophoretic mobility reflects a post-translational modification rather than a gene expression change. Furthermore, as the effect was only very transient (24-48 hours post-transfection) it is not clear that it is relevant for explaining the long term changes in the brain following months of TFEB over-expression. If the goal is to show that TFEB-dependent regulation of autophagy is the relevant mechanism for clearing pTau from the brain, then the measurements should be made in the brain.*

Response: We thank the reviewer for pointing out our erroneous use of language and have corrected the text to address the issue of LC3-II levels as the following (p. 9): "Since ALP can be regulated at both transcriptional and post-transcription levels, we examined whether the autophagy pathway can be activated by TFEB in general. We assessed the steady-state levels

of LC3-II, the best characterized marker of the autophagosome, in TFEB-transfected T40PL cells.”

We agree with the reviewer’s view that the relevance of the transient autophagy induction to the long-term TFEB effect on autophagy in vivo could not be established. Nevertheless, these in vitro results are consistent with and offer mechanistic support for our in vivo findings. Although we have attempted to perform LC3/pTau staining in vivo, we have found it technically challenging to detect LC3 puncta regardless of TFEB state. While this represents an important topic for future study, we feel that comprehensive mechanistic studies in vivo are beyond the scope of the manuscript, a view that the reviewer shared. We have however, provided additional in vivo data for a TFEB regulation of PTEN (new Fig. 8B-E) and lysosomal proteins (new Fig. 6C&D) in this submission.

5. The only evidence to support a change in lysosome abundance/function in the brains that over-express TFEB appears to be the picture of LAMP1 staining in 1 cell with nuclear TFEB in panel 6D. If the major message of the paper is to be that TFEB clears pTau in vivo via a lysosome-dependent mechanism, there needs to be a more robust demonstration that lysosome abundance/function is actually increased in this model. The mRNA studies suggest changes that are consistent with this model but are ultimately meaningless if not robustly corroborated at the protein/organelle level.

Response: Panel 6D was presented as a representative image showing positive correlation between nuclear TFEB with higher LAMP1. This requires high magnification, making documentation of multiple cells difficult. In order to address the reviewer’s concerns we have included additional data showing that TFEB over-expression in mice is associated with higher LAMP1 and CTSD levels (new Figure 6C&D). Although the levels of increases are mild, it is expected considering their important roles in lysosome physiology as well as other lysosomal proteins TFEB regulates. We believe that these together with a general upregulation of lysosomal genes at the transcriptional level by TFEB strengthen a functional role of TFEB in lysosomal function in Tau mice.

6. It is proposed based on transient transfection in cell culture studies that TFEB-dependent up-regulation of PTEN is critical for mediating TFEB-dependent pTau clearance. However, this major point of the study was not confirmed in vivo? Is PTEN up-regulated in the brain in response to TFEB over-expression? Does the degree of PTEN over-expression correlate at the cellular level with pTau clearance? While the results of such experiments would at best be correlative, they would constitute the minimum required to consider the idea that this one out of all the hundreds of TFEB-regulated proteins could have be of such high relevance for the phenomenon under investigation.

Response: As the reviewer requested, we have provided additional in vivo data for a TFEB regulation of PTEN (new Fig. 8B-E), providing correlative support for a role of PTEN in pTau clearance in vivo. We have attempted to perform PTEN/pTau immunostaining and found that anti-PTEN antibodies were not specific enough for us draw any conclusion. We agree with the reviewer that TFEB-mediated clearance of pTau is likely complex, involving many other players in addition to PTEN. Although we have provided compelling evidence for a PTEN-dependent effect of TFEB reduction of pTau, the precise contribution of PTEN in this pathway in vivo requires further investigation.

7. Figure 7 shows that TFEB over-expression promotes up-regulation of PTEN. The degree of TFEB over-expression required to yield this effect should be quantified and shown in order to

helper the reader assess if this is likely to be relevant for the situation of modest TFEB overexpression in vivo.

Response: In the previous version of the manuscript Figure 7 (current Figure 8), we were using the in vitro TFEB overexpression system to verify the potential regulatory connections between TFEB and PTEN expression, in which TFEB is ~20-50 fold overexpressed in the cell culture in different batches of experiments. This level of gross overexpression is expected for transient transfections. TFEB western blots in two different exposures are included upon Referee 2's request (Fig. 8J). One of the advantages of the in vitro system is that we can severely perturb the homeostasis transiently within the cell to dissect out the potential regulatory pathways, which is usually difficult to do in vivo. As mentioned above (see Point 6), we measured PTEN expression levels in the in vivo samples for the current revision, and observed small but statistically significant increase of PTEN in the TFEB-injected brains. This is also expected considering the important function and complex regulation of PTEN in vivo.

8. Was TFEB identified as a PTEN target in the author's previous studies? For example: Sardiello et al, Science, 2009; Palmieri et al, Hum Mol Genet, 2011?

Response: Our data in Figure 7 (now Figure 8) show that PTEN is a novel and direct target of TFEB based on genomic (ChIP followed by qPCR), expression (qPCR and luciferase) and protein (WB of PTEN) analyses. Previous studies from these authors focused on broad-range techniques like ChIP-seq and expression microarray analysis, which are less sensitive than qPCR and luciferase. Hence, albeit ChIP-seq and microarray can help identify putative targets of a transcription factor, focused experiments like the ones performed in Figure 8 can confirm whether or not a candidate gene is a real target of a given transcription factor.

9. Fig. S6. Why is the lysosomal vATPase subunit (Atp6v0d2) whose expression does not change in response to TFEB over-expression grouped with the Autophagy genes rather than the Lysosome genes? If it were grouped with the other lysosomal proteins, the message of this figure would have been greatly diluted as it would suggest that the up-regulation of lysosome genes is not uniform. This raises questions about how the 3 representative examples of lysosome genes (Ctsa, Ctsd, Mcoln1) were selected.

Response: Atp6v0d2 can be grouped as a lysosomal gene, as an autophagy gene, or as neither one according to different systems of classification. Strictly speaking, Atp6v0d2 encodes for a subunit of the V-ATPase, which is assembled on endosomes to acidify them. Only with the fusion with lysosomes, the V-ATPase finds itself associated with the lysosomal membrane. Functionally speaking, the V-ATPase serves both the lysosome and autophagy. Nevertheless, putting Atp6v0d2 in one group or in the other one does not change the reviewer's observation—that the up-regulation of lysosome genes is not uniform, even if all of these genes contain CLEAR sites.

The main concern of the reviewer is: "This raises questions about how the 3 representative examples of lysosome genes (Ctsa, Ctsd, and Mcoln1) were selected." The mRNA examined in Figure S6 is the same used for the global mRNA analysis showed in Fig. 6A. Here, Gene Set Enrichment Analysis shows in an unbiased fashion that the distribution of lysosomal genes is skewed towards upregulated genes (Enrichment Score of all lysosomal genes = 0.56, $P < 0.003$). This result is unbiased and independent of any selection one can make about representative genes. The most important message concerning expression of lysosomal genes is the unbiased data analysis of mRNA presented in Fig. 6A showing that these genes are mostly upregulated upon TFEB overexpression.

Referee #3.

(Remarks):

As commented in the original review "If it could be established firmly that ptau is selectively targeted to an autophagy system, this finding and the functional rescue of the animals by TFEB, are sufficiently exciting with or without a clarification of the targeting mechanism.

The authors have now provided some additional support for the targeting of tau to lysosomes by showing increased PHF1 immunoreactive tau after leupeptin treatment and an example of colocalization of PHF1 immunoreactivity in an LC3-positive compartment. In conjunction with the TFEB effects, the case is reasonable for a role of autophagy in p-tau processing. Proposed explanations for the selective targeting of p-tau are now discussed and the possible involvement of CMA is raised, although the new data showing the colocalization of PHF1 immunoreactivity in an LC3-positive compartment seem to suggest a macroautophagy route. Clearly, however, the mechanism will need to be more thoroughly worked out in a follow-up study. The current data provide impetus for these further investigations. A second earlier concern is whether the TFEB-related changes in autophagy are the sole or main basis for the reduced tauopathy, independently of induced changes in other processes including changes in phosphatases or kinases. In the absence of new data addressing this point, this remains a distinct possibility and the authors have now included statements of caution that acknowledge this possibility.

Response: We appreciate the reviewer's excitement for the novelty and implications of our finding and are pleased that we have successfully addressed most of the reviewer's concerns in our initial revision of the manuscript.

Specific Concerns

Further statements in the response are, however, puzzling : "Our results show that, in all the preparations, while the AT8 (CP13)-positive pTau were reduced, the corresponding unphosphorylated Tau recognized by Tau1 remained constant. This result argues against a shift in phosphorylation" . If not reflecting a change in phosphorylation state, what other process would account for this set of findings? Also for the statement "although it is clear that TFEB targets the detergent-insoluble pTau for degradation, we cannot exclude the possibility that TFEB may affect Tau phosphorylation/dephosphorylation at Tau1-independent sites in the soluble pool" how have the authors excluded autophagy-unrelated effects on detergent-insoluble pTau in addition to any autophagy-related ones? Other earlier concerns have been addressed well with additional data.

Response: We apologize that our results as written were confusing. It was our intent to suggest that the data demonstrate that the reduction in pTau (observed by AT8) as a result of TFEB over-expression is the consequence of pTau being degraded through the autophagy/lysosome pathway rather than through a shift in Tau phosphorylation status. We have clarified the language in this version of the manuscript to read "Since the Tau1 levels remained constant, reduction of AT8-positive Tau most likely indicates that TFEB expression leads to the degradation rather than dephosphorylation of the phospho-Tau (pTau)." (p. 5). We believe this change to the text should address the concerns of the reviewer.

Once again we express our sincere appreciation to the reviewers for their valuable comments.

Thank you for the submission of your revised manuscript to EMBO Molecular Medicine. We appreciate the effort to address all remaining issues. The manuscript was sent back to the most critical referee who sent back her/his report (see below). Unfortunately, as you can see, this referee is still not satisfied and does not support publication of your article in EMBO Molecular Medicine.

Evaluating ourselves the new *in vivo* data provided in Figure 8, we do agree with the referee that the minute up-regulation of PTEN upon TFEB addition questions the biological relevance of the findings and may point towards an additional effect via maybe an undisclosed other TFEB target. Following extensive discussion within the editorial team, and in light of the other two referees' comments, we nevertheless decided to move forward with the manuscript but would suggest moving the PTEN-*in vivo* data to supplementary information and discuss the data accordingly, including moderation of the claim that the mechanism goes via up-regulation of PTEN as the *in vivo* data do not strongly support this.

Please submit your revised manuscript within two weeks.

I look forward to reading a new revised version of your manuscript as soon as possible.

***** Reviewer's comments *****

Referee #2 (Remarks):

The authors have revised the manuscript to better reflect the fact that TFEB over-expression could act through distinct pathways (degradation and/or altered regulation of phosphorylation) to reduce tau-related pathology in a mouse model of Alzheimer's disease. However, new data that has been added to address my previous concerns diminishes the probability that the role for PTEN that was identified as a key downstream target of TFEB in cell culture assays is actually relevant to the *in vivo* effects that arise from TFEB over-expression. I have outlined below the reasoning that leads me to this conclusion.

1. A major limitation of the previous submission was that the authors had not established that modest levels of TFEB over-expression in the mouse brain actually caused changes in PTEN expression that were similar to those observed in the cell culture model. Without such data, it could not be determined whether or not TFEB-dependent changes in tau-linked pathology arise via a PTEN-dependent mechanism. In response to this criticism, the authors now report that they have established that the AAV-based strategy for TFEB over-expression in the mouse brain increases brain PTEN protein levels (Fig. 8B-E). Unfortunately, examination of the anti-PTEN Western blots in 8B and D does not reveal any noticeable change in PTEN protein levels when TFEB is over-expressed. The quantification that accompanies these figures (8C and E) confirms that PTEN levels increase only very minimally (a few percent?) following modest TFEB over-expression. While such minute changes might be statistically significant, it seems improbable that they are physiologically relevant.

Furthermore, it is not clear how the data that is presented in panels 8 C and E was quantified. It is simply said in the methods section that ImageJ was used. The figure legend for 8E states that: "Quantification of relative band intensities of (D). N=3 and 4 per group. Experiment was repeated twice." Why was the experiment performed twice? How were the results from each experiment combined? Was there any normalization of the data? While these questions should have ideally been answered, the fact remains that the changes that were observed are very minimal and are thus of questionable physiological significance.

2. Figure 8J now reveals that the effects on tau metabolism arising from TFEB over-expression in cultured cells arose following the massive over-expression of TFEB. Indeed, in the response to the reviewers, the authors estimate such changes at 20-50 fold. This data should have been quantified in the manuscript. Meanwhile, the extent of TFEB over-expression that was achieved following AAV

injection into the brain was only 2-3 fold (Fig. 8B-E). The fact that the magnitude of TFEB over-expression was so different in cultured cells versus brains raises concerns about whether it is possible to infer that similar mechanisms underlie changes in tau metabolism in these distinct models.

Minor:

In multiple instances, figure legends report that particular experiments were "performed more than 3 times". The exact number of replicates should be reported for each experiment. Since many such experiments were quantified, this should be simple to address.

3rd Revision - authors' response

26 June 2014

We are grateful to the editor and the editorial team for provisionally accepting the manuscript and are pleased to know that we have fully addressed reviewer 1 and 3's comments. Furthermore, we appreciate the editorial team's thoughtful consideration of reviewer 2's most recent comments. In this regard, we have implemented the editorial team's suggestions to move the PTEN *in vivo* data to the supplemental information and to moderate the PTEN based mechanistic claims, which can be found in the abstract, results and discussion sections. Lastly, we have modified the manuscript in accordance with the editorial requirements outlined in comments 1 - 5.

Thank you very much for your decision to move forward with the manuscript.